# Tensile straining of iridium sites in manganese oxides for proton-exchange membrane water electrolysers

Hui Su [1,5] ✉, Chenyu Yang [2,5], Meihuan Liu[3], Xu Zhang[4], Wanlin Zhou[2], Yuhao Zhang[2], Kun Zheng [4], Shixun Lian [1] ✉ & Qinghua Liu [2] ✉

Although the acidic oxygen evolution reaction (OER) plays a crucial role in proton-exchange membrane water electrolysis (PEMWE) devices, challenges remain owing to the lack of efficient and acid-stable electrocatalysts. Herein, we present a low-iridium electrocatalyst in which tensile-strained iridium atoms are localized at manganese-oxide surface cation sites (TS-Ir/MnO$_2$) for high and sustainable OER activity. In situ synchrotron characterizations reveal that the TS-Ir/MnO$_2$ can trigger a continuous localized lattice oxygen-mediated (L-LOM) mechanism. In particular, the L-LOM process could substantially boost the adsorption and transformation of H$_2$O molecules over the oxygen vacancies around the tensile-strained Ir sites and prevent further loss of lattice oxygen atoms in the inner MnO$_2$ bulk to optimize the structural integrity of the catalyst. Importantly, the resultant PEMWE device fabricated using TS-Ir/MnO$_2$ delivers a current density of 500 mA cm$^{-2}$ and operates stably for 200 h.

Proton-exchange membrane water electrolysis (PEMWE) devices, which have lower resistance losses, less gas crossover, and higher current densities than alkaline water electrolysis devices, are a promising and sustainable route for producing clean hydrogen (H$_2$) fuels[1–4]. Presently, the overall efficiency of water electrolysis is mainly limited by the anodic oxygen evolution reaction (OER), which includes a sluggish four proton-coupled electron-transfer process[5–7]. Moreover, severe degradation of active sites in a highly acidic oxidation state, especially at high current densities, severely limits the large-scale deployment of PEMWE devices[8,9]. Although iridium-based metal oxides (IrO$_x$) have been widely used in water electrolysis, they have low mass activities, high overpotentials, and high costs (US\$60,670 kg$_{Ir}^{-1}$) and cannot achieve continuous high activity and durability at high current densities[10,11]. Therefore, although the development of intrinsically active acid-stable low-iridium electrocatalysts that exhibit both enhanced electrocatalytic performance and good long-term durability

is highly desired for promoting more competitive PEMWE devices, it remains a formidable challenge.

The widely-accepted adsorbate evolution mechanism (AEM) is observed for the OER that is catalyzed by iridium oxides and involves multiple intermediates, such as *OH, *O, and *OOH[11]. Usually, the formation of *OOH over metal sites is the rate-limiting step, which is considered as the bottleneck in improving the acidic–OER performance[7,12]. To overcome the scaling relation of the adsorption energies between *OH and *OOH, the catalysts (RuO$_x$) that have strong covalent M–O bonds (where M is a metal atom) can negatively and positively shift the low Hubbard and O 2$p$ bands for metal atoms, respectively, rendering the lattice oxygen atoms more likely to lose electrons via oxidation[13,14]. Therefore, the catalyst follows a lattice oxygen-mediated (LOM) mechanism, which can bypass the production of critical *OOH intermediates for enhanced activity[15,16]. Unfortunately, slow deprotonation at lattice oxygen-atom sites and degradation of

[1]Key Laboratory of Light Energy Conversion Materials of Hunan Province College, College of Chemistry and Chemical Engineering, Hunan Normal University, Changsha 410081 Hunan, China. [2]National Synchrotron Radiation Laboratory, University of Science and Technology of China, Hefei 230029 Anhui, China. [3]State Key Laboratory for Powder Metallurgy, Central South University, Changsha 410083 Hunan, China. [4]Beijing Key Laboratory of Microstructure and Properties of Solids, Faculty of Materials and Manufacturing, Beijing University of Technology, Beijing 100124, China. [5]These authors contributed equally: Hui Su, Chenyu Yang. ✉e-mail: suhui@hunnu.edu.cn; sxlian@hunnu.edu.cn; qhliu@ustc.edu.cn

metal species on the catalyst surface preclude long-term operating stability[17,18]. In particular, the oxidative release of lattice oxygen atoms can overoxidize metal species (such as Ru) to dissolvable high-valence metal oxides ($RuO_4$) during acidic−OER processes. Moreover, violent reactions involving lattice oxygen atoms produce numerous oxygen vacancies, which directly dissolve metals on the catalyst surface[19]. Even worse, during prolonged reactions, easily formed oxygen vacancies deeply penetrate the inner bulk of the material, which finally collapses the material structure and thus deactivates the catalyst.

Recently, $MnO_2$ is an abundant and inexpensive transition metal oxide with high surface activity and catalytic stability in acidic media, which is considered as a potential carrier for acidic−OER reactions. The $IrO_2$ nanoparticles and a low loading of 5% iridium supported on the α-$MnO_2$ electrocatalyst exhibited good electrocatalytic OER performance due to high active specific surface areas and more high-valence-state iridium[20,21]. Liu's group presented a light-driven strategy to realize an orderly Ir atomic assembly on a F doped $MnO_2$ (IrMnOF) surface with a spin-related lower entropy that is optimized to reduce the intrinsic activation energy at potential-determining intermediates for a stable OER process[22]. To break the linear relationship of multiple reaction intermediates for increasing activity, Ru-atom-array patches supported on α-$MnO_2$ with appropriate atomic distances in symmetric dual-metal sites delivered enhanced acidic−OER activity following an oxide path mechanism[11]. Furthermore, a strained lattice containing optimized Ir−O−Ir and Ir−O−Mn bonding along with the presence of $Mn^{3+}$ in the high-loading 22 wt% Ir-incorporated β-$MnO_2$ exhibited enhanced OER stability[23]. Therefore, simultaneously breaking the linear relationship of multiple reaction intermediates and inhibiting a large number of oxygen vacancies is essentially desirable for designing advanced electrocatalysts, but it is a great challenge.

Therefore, in this study, we used cation exchange and a subsequent rapid annealing−cooling strategy to prepare a low-iridium electrocatalyst in which iridium atoms are localized at the surface Mn sites of tensile-strained manganese oxide (TS−Ir/$MnO_2$). The tensile strain introduced by the twisted square planar (Ir−$O_4$) moieties confined in the acid-resistant $MnO_2$ can enhance the covalency of the Ir−O bond to improve the deprotonation ability and increase the bond's orbital overlap degree with the metal $d$ band during the reaction[13,24–26]. In particular, the tensile strain localized on the $MnO_2$ surface could tailor the adsorption behavior of Ir sites to accelerate the deprotonation of *OH at surface oxygen vacancies and, thus, effectively prevent the local peroxidation of Ir sites to reduce the dissolution and maintain the structural integrity of the catalyst. Consequently, the acidic−OER that occurred on the TS−Ir/$MnO_2$ surface followed a continuous localized lattice oxygen-mediated (L-LOM) mechanism, which causes the catalyst to deliver a high mass activity of 1025 A $g_{Ir}^{-1}$ at an overpotential of 198 mV (typically at a current density of 10 mA $cm^{-2}$), which is approximately 19 and 380 times higher than those of Ir−$MnO_2$ (54 A $g_{Ir}^{-1}$) and commercial $IrO_2$ (2.7 A $g_{Ir}^{-1}$), respectively. The L-LOM mechanism is experimentally elucidated using in situ X-ray absorption fine structure (XAFS) and synchrotron radiation infrared (SRIR) spectroscopic measurements. Most importantly, in situ isotope-labeling SRIR measurements confirm the rapid adsorption of $H_2O$ molecules on surface oxygen vacancies and the rapid deprotonation of *OH over lattice oxygen atoms to trigger a continuous L-LOM catalytic reaction for stabilizing surface Ir active sites. Therefore, the TS−Ir/$MnO_2$ catalyst exhibits stable acidic−water electrolysis of 100 and 200 h at current densities of 200 and 500 mA $cm^{-2}$ in a three-electrode system and the resultant PEMWE device, respectively, with negligible performance degradation, which suggests good potential for practical application in PEMWEs.

## Results and discussion

### Synthesis and characterization of catalysts

A low-iridium electrocatalyst in which atomically dispersed and tensile-strained iridium sites were confined in manganese oxides

(TS−Ir/$MnO_2$) was hydrothermally synthesized and then rapidly thermally annealed and subsequently cooled, which is hereafter called the "rapid thermal annealing−cooling" (RTAC) strategy. Scanning electron microscopy (SEM) and transmission electron microscopy (TEM) images show the $MnO_2$ nanofiber morphology, which exposes more open metal sites (Supplementary Information Fig. 1). $MnO_2$ nanofibers were transferred to an aqueous $IrCl_3$ solution for the cation-exchange reaction. The as-obtained sample was rapidly heated to 250 °C and then rapidly cooled to introduce tensile strain to the nanofibers. Subsequent SEM (Fig. 1a) and TEM (Fig. 1b) images clearly reveal that the ultrathin nanofiber morphology was retained after the Ir exchange reaction and rapid pyrolysis for the TS−Ir/$MnO_2$ sample. As shown in Fig. 1c and Supplementary Fig. 2, the X-ray diffraction (XRD) patterns for the TS−Ir/$MnO_2$ sample present typical characteristic diffraction peaks of α-$MnO_2$ without any peaks attributed to Ir-related phases. After Ir doping and subsequent RTAC, the $2\theta$ peaks slightly shifted toward lower angles, which implies that tensile strain formed in the sample. Furthermore, the XRD results of the Ir samples with different Ir doping concentrations and annealing methods clearly reveal that a rapid annealing−cooling treatment introduces tensile strain in the host $MnO_2$ lattice resulting in a larger shift in the XRD pattern, excluding the effect of low loading Ir atoms on the lattice of host $MnO_2$.

Moreover, as shown in the high-resolution TEM (HRTEM) images in Fig. 1d–f, the fringe lattice parameter of the TS−Ir/$MnO_2$ nanofibers was measured at 2.41 Å, which corresponds to the $MnO_2$ (211) planes[27]. For comparison, Ir atoms were dispersed on the tensile strain−free $MnO_2$ nanofiber surface via slow annealing and subsequent cooling, for which the TEM and HRTEM images (Supplementary Figs. 3 and 4, respectively) show that the fringe lattice parameter is 2.36 Å. The increased lattice spacing represents the RTAC-driven tensile strain generated on the TS−Ir/$MnO_2$ nanofiber surface. To further clarify the morphological structure of Ir sites, the elemental mappings (Supplementary Fig. 5) indicate the uniform distribution of Ir, Mn, and O in the nanofiber. Similarly, Ir, Mn, and O are uniformly distributed in Ir−$MnO_2$ (Supplementary Fig. 6). Aberration-corrected high-angle annular dark-field scanning TEM (HAADF-STEM, Fig. 1g, h) images clearly show Ir atomically dispersed in the obtained TS−Ir/$MnO_2$ nanofibers. The Ir atoms, highlighted by scattered bright dots in the lattice, are at the same locations as columns of Mn atoms (Fig. 1i), which suggests that atomically dispersed Ir replaces surface Mn sites in the $MnO_2$ nanofiber lattice via a cation-exchange reaction. In this RTAC process, the rapid annealing treatment introduces stress in the lattice, and then rapid cooling does not undergo a slow cooling process for stress release. During this process, the tensile strain was retained in TS−Ir/$MnO_2$. Similarly, Ir is also atomically dispersed over the $MnO_2$ nanofibers in Ir−$MnO_2$ (Supplementary Fig. 7). The Ir loading obtained using inductively coupled plasma optical emission spectroscopy (ICP−OES) is approximately 3.9 wt% in TS−Ir/$MnO_2$. Furthermore, the ICP−OES result shows that the Ir loading is similar in Ir−$MnO_2$, which is attributed to the same raw material feeding. These results confirm the uniform dispersion of Ir atoms on the surface Mn sites of $MnO_2$ nanofibers and that the tensile strain of Ir sites was introduced to TS−Ir/$MnO_2$.

### Electronic properties of catalysts

To clarify the local electronic structure and binding environment of Ir species in the TS−Ir/$MnO_2$ nanofibers, X-ray photoelectron spectroscopy (XPS) and X-ray absorption spectroscopy (XAS) were performed. The Ir 4$f$ XPS spectrum in Fig. 2a shows an obvious peak at 65.36 eV (Ir 4$f_{5/2}$), which exhibits a 0.59 eV positively shifted binding energy compared to that of $IrO_2$ (64.77 eV) and suggests that the increased oxidation state of Ir could be attributed to the electronic interaction between Ir and Mn[28]. Correspondingly, the Mn 2$p_{3/2}$ XPS spectrum in Supplementary Fig. 8 shows a negative-energy shift after the introduction of Ir atoms, which reveals the electron transfer from Ir to Mn. Importantly, RTAC can further increase the electron transfer to

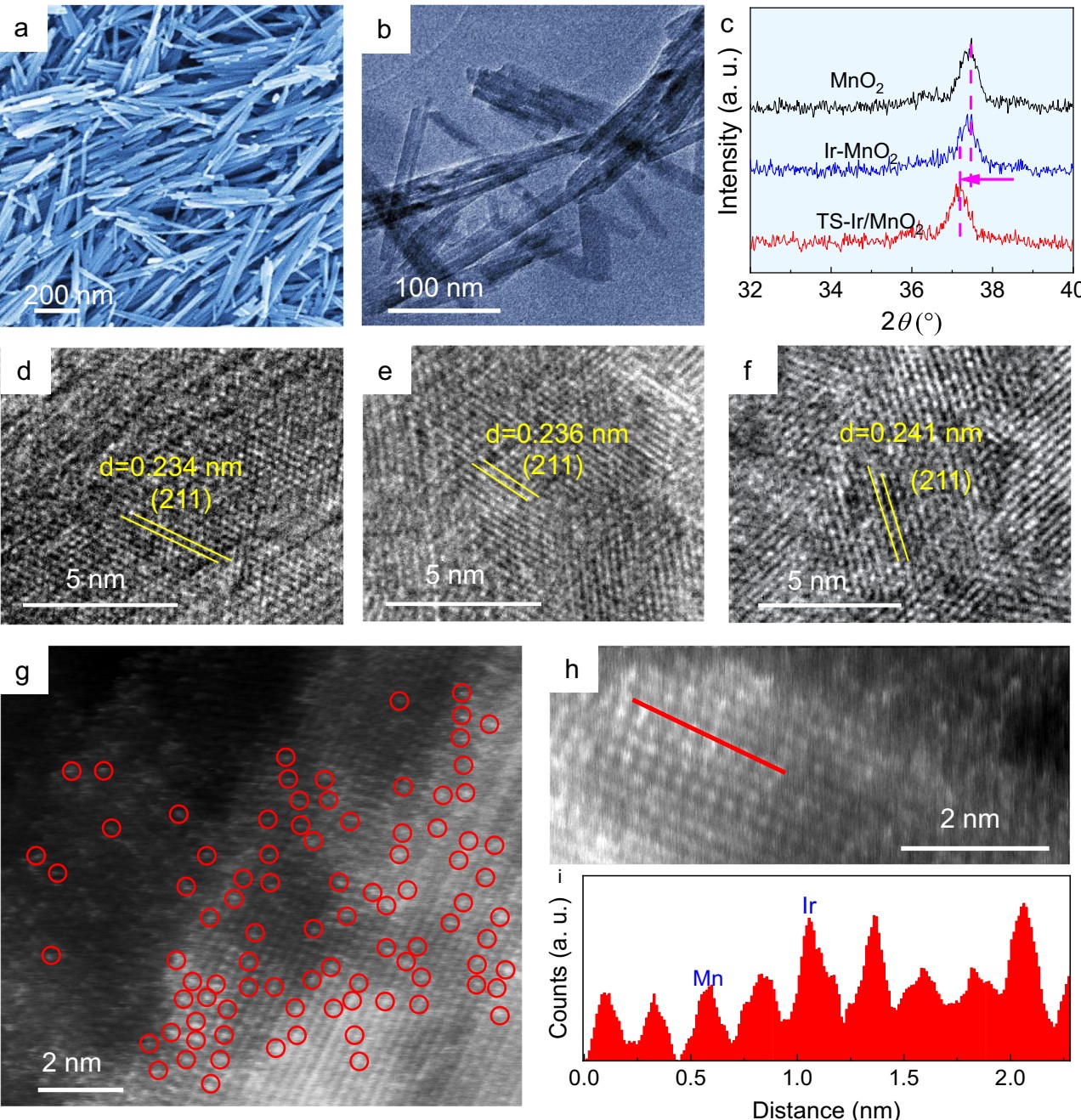

**Fig. 1 | Structural characterization of the TS–Ir/MnO₂ electrocatalyst. a** SEM and **b** TEM images of the TS–Ir/MnO₂ electrocatalyst. **c** XRD patterns and **d–f** HRTEM images of MnO₂, Ir–MnO₂, and TS–Ir/MnO₂ electrocatalysts. **g** HAADF–TEM image, in which bright spots highlighted by red circles are ascribed to Ir sites. **h** Enlarged HAADF–TEM image and **i** line profiles for HAADF intensity analysis labeled in **h** for the TS–Ir/MnO₂ electrocatalyst.

increase the oxidation state of Ir species in the TS–Ir/MnO₂ nanofibers. The increased Ir valence state can be further verified by Ir and Mn XAS spectra[29,30]. The X-ray absorption near-edge spectroscopy (XANES) spectra of the Ir $L_3$-edge in Fig. 2b show that the white-line peak for TS–Ir/MnO₂ is more intense than those for both IrO₂ and Ir–MnO₂, which indicates that the valence state slightly increased. Interestingly, the XANES spectrum of the Mn $K$-edge shows the opposite trend; that is, the white-line peak for Mn weakens, which means that the oxidation state of Mn decreases after the introduction of Ir species and tensile strain (Fig. 2c). These results are consistent with the XPS spectra of Ir and Mn, which indicates the electron transfer from Ir to Mn.

Furthermore, extended XAFS (EXAFS) spectra reveals the local coordination environments of Ir species in the samples[31]. In the

Fourier-transform (FT) curves of the Ir $L_3$-edge EXAFS spectra, the main peaks in the range of 1.5–1.6 Å can be assigned to the first shell of the Ir–O coordination. Additionally, the O 1s XPS spectra support the existence of Ir–O bonds in the sample (Supplementary Fig. 9). Furthermore, the higher binding energy of Ir–O in TS–Ir/MnO₂ further demonstrates that the increased oxidation state of Ir due to electron transfer driven by tensile strain. A slight difference in the position of the main peak at approximately 1.5 Å in Fig. 2d represents a different apparent Ir–O bond length in TS–Ir/MnO₂, Ir–MnO₂, and IrO₂. Figure 2d, Supplementary Fig. 10 and Supplementary Table 1 show that the dominant coordination peak at 1.48 Å, which is assigned to the first shell of Ir–O in Ir–MnO₂, is shorter than that at 1.60 Å (for Ir–O bonds in IrO₂), which is attributed to the distinct second shell structure between

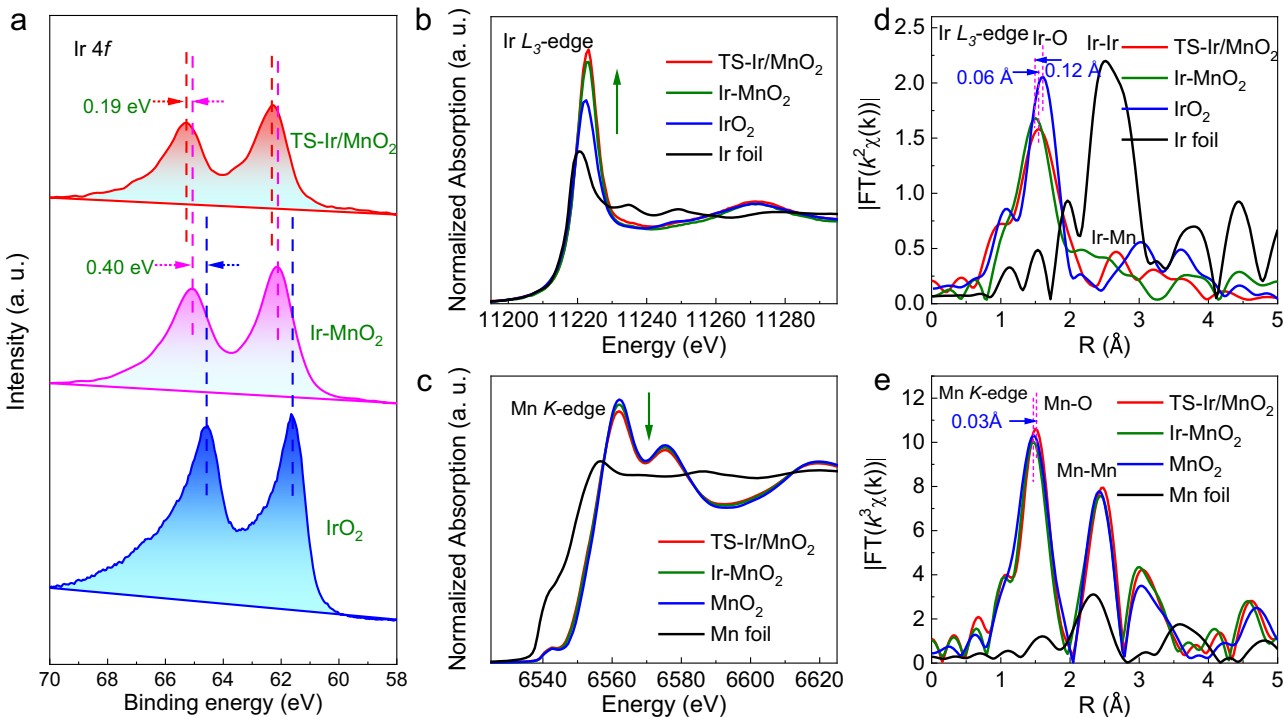

**Fig. 2 | Electronic property characterizations of electrocatalysts. a** Ir 4*f* XPS spectra of TS–Ir/MnO₂, Ir–MnO₂, and IrO₂. **b** Ir *L*₃-edge XANES spectra and **d** Fourier transforms (FTs) of Ir *L*₃-edge EXAFS oscillations for TS–Ir/MnO₂, Ir–MnO₂, IrO₂, and Ir foil. **c** Mn *K*-edge XANES spectra and **e** FTs of Mn *K*-edge EXAFS oscillations for TS–Ir/MnO₂, Ir–MnO₂, MnO₂, and Mn foil.

Ir–MnO₂ and IrO₂. Moreover, the apparent bond length of the Ir–O first shell in TS–Ir/MnO₂ (1.54 Å) is slightly longer than that of the Ir–O first shell in Ir–MnO₂, which is attributed to the tensile strain. Meanwhile, the Ir–O bond length in TS–Ir/MnO₂ (1.54 Å) is shorter than that in IrO₂ (1.60 Å), which suggests that the increased covalency and electron transfer of the Ir–O bond were suitable in TS–Ir/MnO₂. In order to indeed evaluate the real bond length of the Ir–O coordination, the fitted curves of the $k^2$-weighted Ir *L*₃-edge EXAFS spectra of all samples in Supplementary Fig. 11, for which the corresponding data are listed in Supplementary Table 2, show that the coordination number of the Ir–O first shell is ~4 and that the real Ir–O bond length is 1.94 Å in TS–Ir/MnO₂, which is shorter than that in IrO₂ (2.01 Å) and longer than that in Ir–MnO₂ (1.89 Å), suggesting the existence of tensile strain and increased covalency. Furthermore, after Ir was introduced to MnO₂, the FT-EXAFS spectra of the Mn *K*-edge in Fig. 2e shows no obvious change in the bond length of the Mn–O first shell, which is primarily due to the low Ir content. However, the tensile strain effect could be found by comparing the Mn–O bond lengths in TS–Ir/MnO₂ versus MnO₂ (Supplementary Table 1), and the Mn–O bond lengths increased by 0.03 Å, suggesting that tensile stress existed in host MnO₂ for TS–Ir/MnO₂. However, the tensile strain effect could also be found by comparing the Mn–O bond lengths in TS–Ir/MnO₂ vs. MnO₂, which reveals the origin of the Ir-site tensile strain in the MnO₂ nanofibers. Furthermore, another peak at 2.66 Å is shorter than at 3.03 Å (for Ir–Ir bond at IrO₂) and is larger than at 2.45 Å (for Mn–Mn bond at MnO₂), suggesting that the peak can be assigned to Ir–Mn at TS–Ir/MnO₂. This further demonstrates that atomically dispersed Ir replaces surface Mn sites in the MnO₂ nanofiber lattice via a cation-exchange reaction in TS–Ir/MnO₂. Therefore, these results reveal that the elongation of surface Ir–O bonds in TS–Ir/MnO₂ effectively tunes the covalency and electronic structures of Ir sites in the samples.

## Electrochemical characterization in acidic electrolyte

To evaluate the electrocatalytic OER activity of TS–Ir/MnO₂, a three-electrode electrochemical workstation was used for measurements in a 0.1 M HClO₄ electrolyte, and tensile strain–free Ir–MnO₂, MnO₂, and IrO₂ were employed as reference samples. For an increased number of opening sites and accessible species, carbon cloth was selected to support catalysts for electrochemical measurements (Supplementary Fig. 12). The linear sweep voltammetry (LSV) curves of the TS–Ir/MnO₂ and reference samples are shown in Fig. 3a. To achieve a typical current density of 10 mA cm⁻², TS–Ir/MnO₂ only requires an ultralow overpotential ($\eta$) of 198 mV, which is lower than those of Ir–MnO₂ (275 mV), MnO₂ (396 mV), and commercial IrO₂ (316 mV). Moreover, catalysts were prepared with different Ir loadings, among which TS–Ir/MnO₂ with a moderate Ir loading (3.9 wt%) exhibited the best OER activity (Supplementary Fig. 13). Interestingly, TS–Ir/MnO₂ exhibits the best acidic activity at a high current density of 200 mA cm⁻² and only requires a low overpotential of 356 mV (Fig. 3b). The morphology structure and OER performance of Ir electrocatalysts at different annealing pyrolysis temperatures are shown in Supplementary Fig. 14, revealing the optimal tensile strain and atomically dispersed Ir active sites in the TS–Ir/MnO₂ (250 °C) electrocatalyst. Considering the effect of temperature on acidic-OER performance, we conducted a water splitting test under different operating temperatures. The OER activity is quite temperature dependent, and the overpotentials of the TS–Ir/MnO₂ electrocatalyst decrease from 198 to 180 mV at a current density of 10 mA cm⁻² with the increase of temperature to 80 °C, which is consistent with what has been observed that a higher temperature provides faster OER kinetics and better OER activity (Supplementary Fig. 15). Moreover, the acidic-OER kinetics of TS–Ir/MnO₂ was analyzed, as shown in Fig. 3c. Clearly, TS–Ir/MnO₂ exhibits the gentlest Tafel slope of 56.6 mV dec⁻¹, which is substantially gentler than those of Ir–MnO₂ (101.2 mV dec⁻¹), MnO₂ (264.7 mV dec⁻¹), and commercial IrO₂ (76.1 mV dec⁻¹) and reveals faster OER kinetics and electron transfer over Ir active sites in TS–Ir/MnO₂ owing to suitably covalent Ir–O bonds. TS–Ir/MnO₂ delivers an efficient catalytic kinetic activity in Supplementary Table 3, which is beyond that of comparable OER catalysts. Furthermore, according to electrochemical impedance spectroscopy analysis, TS–Ir/MnO₂ has negligible charge-transfer

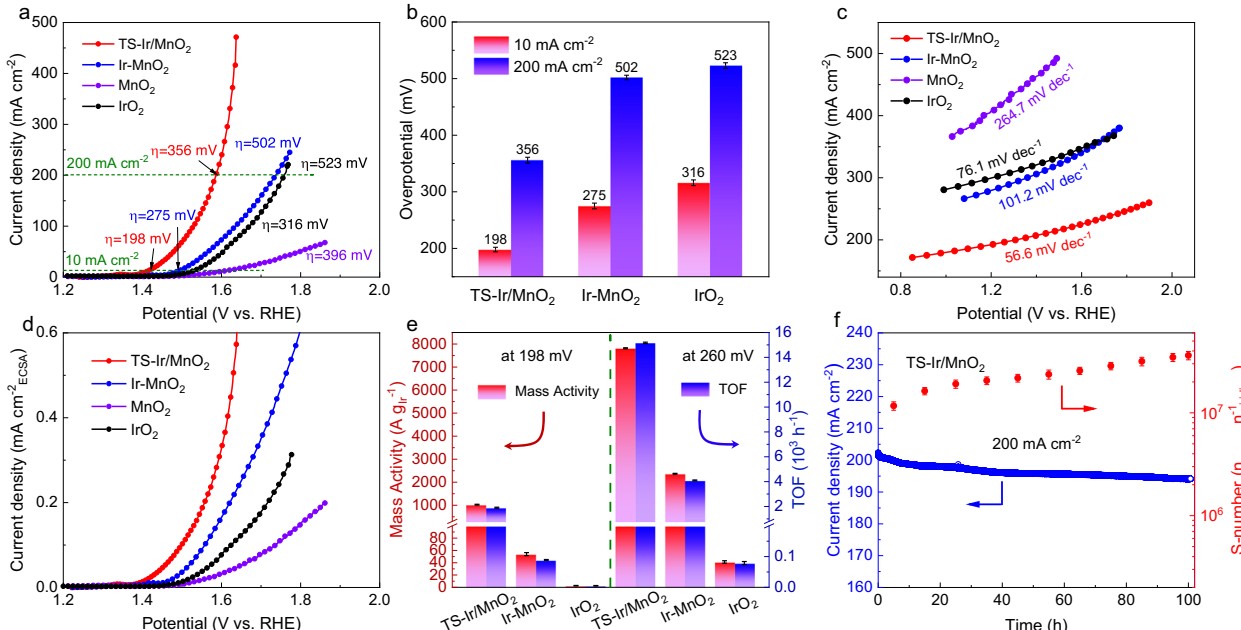

**Fig. 3 | Electrocatalytic OER properties of electrocatalysts in 0.1 M HClO$_4$ (pH = 1). a** Linear sweep voltammetry (LSV) curves, **b** overpotentials at 10 and 200 mA cm$^{-2}$ (error bars are the standard deviations of three replicate calculation). The loading of catalysts is 0.25 mg cm$^{-2}$, and the solution resistance is 2.2 Ω. **c** Tafel slopes, and **d** OER polarization curves based on electrochemical surface area (ECSA) for TS−Ir/MnO$_2$, Ir−MnO$_2$, MnO$_2$, and IrO$_2$ supported on carbon cloths. **e** Mass activity and turnover frequency (TOF) at overpotentials of 198 and 260 mV for TS−Ir/MnO$_2$, Ir−MnO$_2$, and IrO$_2$. Error bars are the standard deviations of three replicate calculation. **f** OER stability tests and constant S-numbers calculated at a constant potential of 1.60 V (driving a current density of 200 mA cm$^{-2}$) for TS−Ir/MnO$_2$.

resistance and therefore the fastest reaction kinetics (Supplementary Fig. 16).

Importantly, the electrochemically active surface area (ECSA) of TS−Ir/MnO$_2$ was achieved via the roughness factor, and double-layer capacitances ($C_{dl}$) were measured (Supplementary Figs. 17 and 18 and Table 4)[32]. Notably, TS−Ir/MnO$_2$ has the highest $C_{dl}$ of 23.62 mF cm$^{-2}$, which suggests more open active sites to achieve high-efficiency OER activity. As expected, Fig. 3d shows that TS−Ir/MnO$_2$ also delivers the best specific activity when the current density is normalized per the ECSA. These results reveal the superior acidic−OER activity of TS−Ir/MnO$_2$. To further assess the intrinsic activity of TS−Ir/MnO$_2$, the mass activity and turnover frequency (TOF) were calculated based on the loading of Ir sites, as shown in Fig. 3e. TS−Ir/MnO$_2$ has a high mass activity of 1025 A g$_{Ir}^{-1}$ and an ultrahigh TOF of 1900 h$^{-1}$ at a typical overpotential of 198 mV (reaching a current density of 10 mA cm$^{-2}$), which is 380 times higher than those of commercial IrO$_2$ (2.7 A g$_{Ir}^{-1}$ and 5 h$^{-1}$, respectively). With increasing working potentials, both the mass activity and TOF of TS−Ir/MnO$_2$ substantially increase compared to those of IrO$_2$, which confirms that TS−Ir/MnO$_2$ has faster OER kinetics than IrO$_2$. More importantly, TS−Ir/MnO$_2$ achieved an ultrahigh mass activity of 51250 A g$_{Ir}^{-1}$ at 1.65 V (Supplementary Table 5), which is much higher than the target (6000 A g$_{Ir}^{-1}$ at 1.7 V) set by the International Renewable Energy Agency (IRENA). In other words, this is equivalent to an 85% reduction in the amount of the precious metal (Ir) that is used to achieve a similar current power for practical applications.

Moreover, operation durability is very important for practical applications of electrocatalysts. To evaluate the operational durability of the catalyst, OER polarization curves (Supplementary Fig. 19) and chronoamperometry (Supplementary Fig. 21) were measured at typical potentials. TS−Ir/MnO$_2$ presents good catalytic stability at a current density of 10 mA cm$^{-2}$ after 200 h of continuous OER tests with negligible performance degradation. To further clarify the electrochemical durability at higher current densities, continuous OER tests were

performed for TS−Ir/MnO$_2$ at a current density of 200 mA cm$^{-2}$, as shown in Fig. 3f. TS−Ir/MnO$_2$ still maintains a satisfactory ~93% of the initial current density after 100 h of operation. Meanwhile, only ~4.8% of the Ir leached from TS−Ir/MnO$_2$ after continuous OER operation, which is attributed to the strong lattice-confined effect of the robust MnO$_2$ substrate for inhibiting the dissolution of Ir active sites. As shown in Supplementary Fig. 21, TS−Ir/MnO$_2$ has a high S-number (~10$^7$ $n_{oxygen}$ $n_{Ir}^{-1}$) at 200 mA cm$^{-2}$, which suggests good acidic−OER stability[33]. The Ir concentration underwent a slow increase and no decreasing trend was observed during the first few hours of OER operation, suggesting that the dissolved Ir did not have an obvious dynamic sediment-dissolution equilibrium. Moreover, morphological and structural characterizations, such as TEM, XPS, and XAFS (Supplementary Figs. 22−24, respectively), after long-term electrochemical measurements clearly show that Ir is atomically dispersed in MnO$_2$ without agglomeration and that the Ir oxidation state and local coordination structure negligibly change. Notably, the lattice oxygen atoms show a slightly positive shift after the OER measurements, which reveals that the oxidized lattice oxygen atoms can further participate in the catalytic reaction (Supplementary Fig. 25). These results confirm the good stability of the TS−Ir/MnO$_2$ catalyst during continuous acidic−OER operation. In summary, the high durability of the TS−Ir/MnO$_2$ electrocatalyst under continuous operating conditions is attributed to the stable structure and a continuous L-LOM mechanism during the OER process, which can inhibit peroxidation and dissolution of Ir active sites.

## Exploration of mechanism via in situ characterization

To elucidate the catalytic mechanism, in situ SRIR and XAFS spectra were measured using homemade cells under typical potential conditions[25,34−36]. As shown in Fig. 4a, the in situ SRIR spectra show no obvious absorption bands for TS−Ir/MnO$_2$ in the range 700−1700 cm$^{-1}$ at potentials <1.15 V vs. a reversible hydrogen electrode (RHE, the potentials mentioned below are all relative to RHE). With increasing

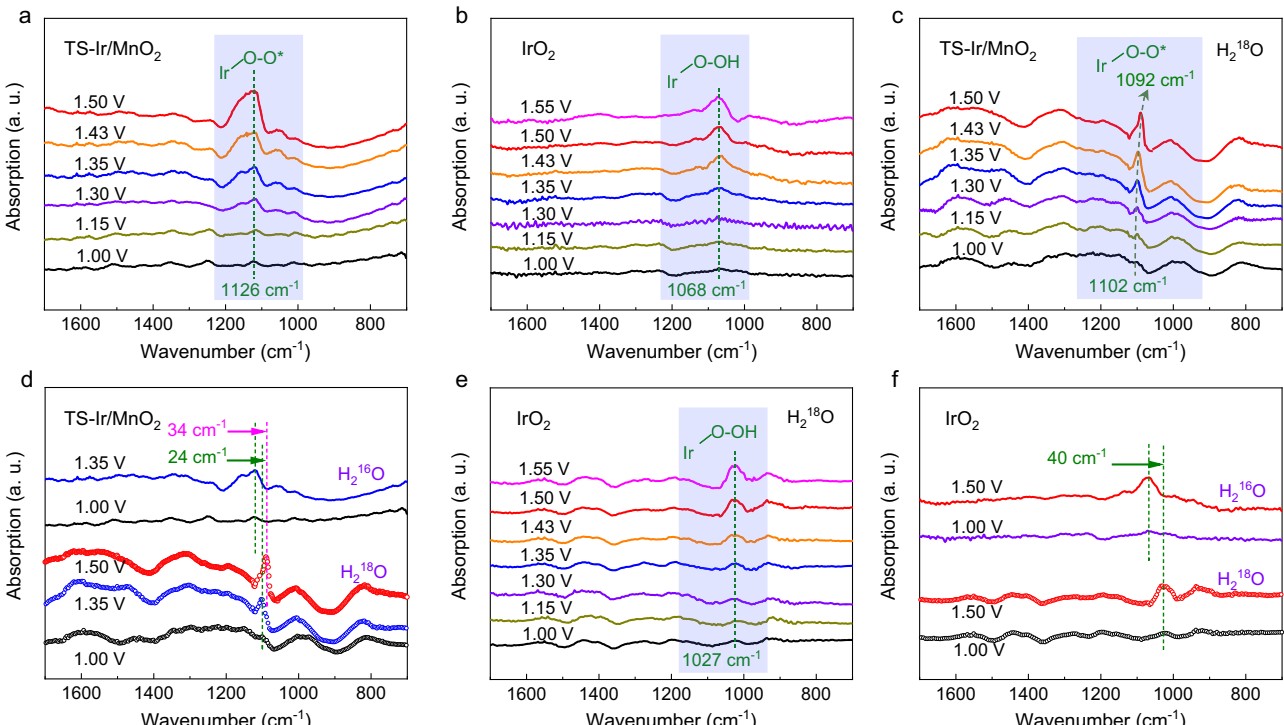

**Fig. 4 | In situ SRIR measurements in 0.1 M HClO₄ (pH = 1). The loading of catalysts is 1 mg cm⁻², and the solution resistance 2.2 Ω.** In situ SRIR spectroscopy measurements in the range 1700–700 cm⁻¹ at various potentials for **a** TS–Ir/MnO₂ and **b** IrO₂. Isotope-labeling in situ SRIR spectroscopy measurements in the range 1700–700 cm⁻¹ at various potentials for **c** TS–Ir/MnO₂ and **e** IrO₂. **d** In situ SRIR spectra generated under no- and isotope-labeling conditions at typical potentials of 1.00 and 1.35 V for TS–Ir/MnO₂. **f** In situ SRIR spectra generated under no- and isotope-labeling conditions at typical potentials of 1.00 and 1.50 V for IrO₂.

potential to 1.30 V vs. RHE, an absorption band appeared at 1126 cm⁻¹ and could be assigned to the key O–O intermediates because the oxygen species (O–O) stretching vibration is usually in the range 1100–1150 cm⁻¹ [37,38]. Interestingly, the intensity of this absorption band is positively correlated with the applied potential, which reveals the rapid accumulation of key O–O radical intermediates over the active sites. For comparison, in situ SRIR spectra were measured under similar OER conditions for IrO₂, as shown in Fig. 4b. With gradually increasing applied potential, an IR absorption band appeared at 1068 cm⁻¹. Because the infrared vibration peaks of the *OOH species usually appear in the region 1000–1100 cm⁻¹, the absorption band at 1068 cm⁻¹ can be assigned to the production of *OOH species over the Ir sites [39,40]. The SRIR result of Ir–MnO₂ further shows the production of key *OOH species under working conditions, suggesting that Ir–MnO₂ catalyses the OER that follows a kinetic-slow AEM pathway (Supplementary Fig. 26). Notably, during the OER, only O–O radicals accumulated for TS–Ir/MnO₂, which bypassed the formation of the sluggish *OOH species. These results suggest that on the TS–Ir/MnO₂ surface, the OER undergoes a different reaction mechanism rather than the conventional AEM on the IrO₂ surface.

To confirm this hypothesis, isotope-labeling in situ SRIR spectra were measured for TS–Ir/MnO₂ and IrO₂ with increasing applied potential from 1.00 to 1.55 V vs. RHE. Usually, for oxygen species (O–O), the ¹⁶O/¹⁸O isotope exchange redshifts the vibrational band by ~20 cm⁻¹. Correspondingly, for oxygen species (O–O), the ¹⁸O/¹⁸O isotope exchange can redshift the vibrational band by approximately 30–40 cm⁻¹ [25,41]. As shown in Fig. 4c, under working conditions, the dominant absorption band appeared at 1102 cm⁻¹ for TS–Ir/MnO₂, and the band's intensification indicates potential dependence tolerance. More interestingly, with increasing applied potential to 1.50 V, the dominant O–O absorption band redshifts, which reveals that the ¹⁸O (O–O) proportion gradually increases with increasing applied potential. At an applied potential of 1.35 V, the absorption band redshifted

from 1126 to 1102 cm⁻¹ when the H₂¹⁶O solution was replaced by H₂¹⁸O, which confirmed that during the OER, ¹⁶O–¹⁸O radicals are derived from adsorbed H₂O molecules and lattice oxygen atoms (Fig. 4d). These results reveal an LOM-like mechanism on the TS–Ir/MnO₂ catalyst surface. Moreover, with increasing applied potential to 1.50 V, the absorption band further redshifted to 1092 cm⁻¹, which suggests that water molecules rapidly fill oxygen vacancies to continuously trigger the LOM-like mechanism. Meanwhile, for IrO₂ (Fig. 4e, f), the *OOH absorption band appeared at 1027 cm⁻¹ and gradually intensified with increasing applied potentials. Notably, the *OOH absorption band shifts from 1068 to 1027 cm⁻¹ when H₂¹⁶O transforms to H₂¹⁸O, which confirms that emerging ¹⁸O–¹⁸OH intermediate species are derived from H₂O molecules adsorbed at active sites. These results confirm that the OER on the TS–Ir/MnO₂ surface follows a different catalytic reaction path than that on the surface of commercial IrO₂ and bypasses the *OOH intermediate during the reaction to accelerate the four-electron reaction path.

To elucidate the underlying mechanism for the TS–Ir/MnO₂ catalyst, in situ XAFS spectra, which are sensitive to the local structural evolution, were measured using a homemade cell fabricated based on the three-electrode system [42,43]. Fig. 5a shows the in situ XANES spectra of the Ir $L_3$-edge for TS–Ir/MnO₂ recorded at different applied potentials. Compared with the ex situ state (immersion in an acidic solution without applying any potential), the white-line peak slightly intensifies and positively shifts with increasing applied potential to 1.43 V vs. RHE, which suggests that the oxidation state of Ir sites is elevated. This is attributed to more electrons moving from Ir to nearby adsorbed oxygen species and then promoting a rapid oxidation reaction. The FT-EXAFS spectra of the Ir $L_3$-edge (Fig. 5b and Supplementary Fig. 27) show a dominant peak at ~1.55 Å, which is assigned to the first shell of the Ir–O bond. Compared with the ex situ state, the first-shell peak intensified at an applied potential of 1.15 V and exhibited a slight high-R shift from 1.54 to 1.59 Å (Supplementary Table 6). With increasing

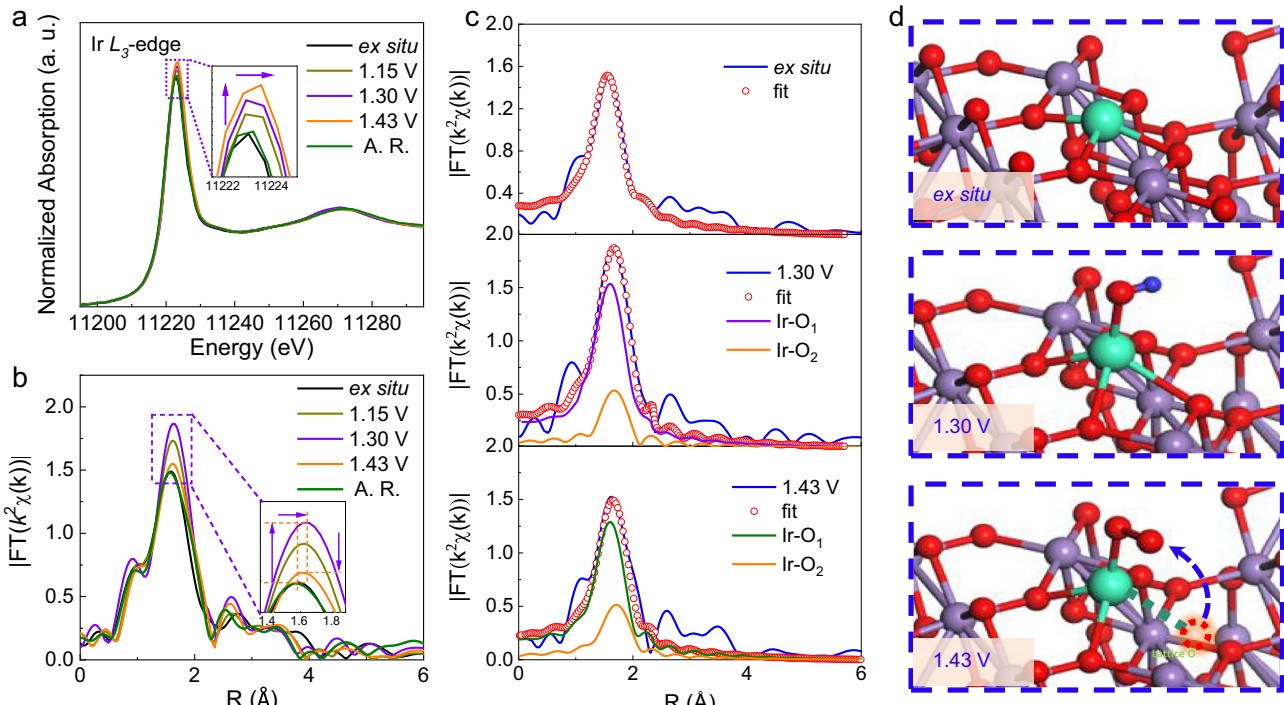

**Fig. 5 | In situ XAFS measurements in 0.1 M HClO₄ (pH = 1). The loading of catalysts is 1 mg cm⁻², and the solution resistance 2.2 Ω. a** In situ XANES spectra of Ir $L_3$-edge for TS-Ir/MnO₂ at typical potentials during OER. **b** Corresponding $k^2$- weighted FT-XAFS spectra. **c** Fitted curves and **d** OER mechanism diagrams of TS-Ir/MnO₂ under ex situ, 1.30 and 1.43 V conditions. The inset shows Mn, O, Ir, and H atoms indicated by purple, red, green, and royal blue, respectively.

applied potential to 1.30 V, the Ir–O coordination peak further intensified. These results imply the rearrangement of the local coordination structure of Ir sites during the OER, which is due to the adsorption of oxygen species over Ir sites. Interestingly, when a typical potential of 1.43 V is applied to the electrode, the coordination peak weakens, which suggests that the Ir coordination number is reduced under OER conditions (oxygen vacancy generation). Moreover, the in situ Mn $K$-edge XAFS spectra further reveal that the bulk phase material maintains its structural stability (Supplementary Fig. 28). Most importantly, after long-term OER measurements for TS-Ir/MnO₂, the Ir coordination number approximated that under ex situ conditions. The infrared spectrum of the isotopically labeled sample after the reaction shown in Supplementary Fig. 29 reveals that the peak for the M–O (M = Ir and Mn) band redshifts by -13 cm⁻¹, which confirms that TS-Ir/MnO₂ contains ¹⁸O after the reaction. These results confirm that localized oxygen vacancies on the TS–Ir/MnO₂ surface were rapidly filled by H₂O molecules to modify the LOM mechanism, which is named the "localized LOM" (L-LOM) mechanism hereafter.

The FT-XAFS fitted curves are shown in Fig. 5c and Supplementary Figs. 30 and 31, and the corresponding data are listed in Supplementary Table 7 for the configuration of four Ir–O₄ coordination bonds that are quantitatively optimized ex situ. At 1.30 V, an additional first shell was considered for the longer Ir–O coordination bond (2.08 Å), which implies that oxygen species adsorbed on Ir sites. With increasing applied potential to 1.43 V, the coordination number of the shorter Ir–O bonds reduced to three ($R$ = 1.96 Å), and one longer Ir–O coordination bond appeared at $R$ = 2.08 Å. These results imply that for the tensile-strained Ir sites, a potential-driven structural evolution occurs during the OER by releasing lattice oxygen atoms from TS-Ir/MnO₂ localized surface sites to induce the L-LOM reaction mechanism. The comprehensive analysis of the fitted curves obtained for Ir sites and O–O radicals probed via in situ SRIR suggests that oxygen molecules that adsorb on tensile-strained Ir sites can be quickly deprotonated and coupled with adjacent lattice oxygen atoms to form O–O radicals

as *OO–Ir–O₃, which bypasses sluggish *OOH intermediates (Fig. 5d). The proposed L-LOM mechanism can rapidly accelerate deprotonated oxygen–containing intermediates and then couple lattice oxygen atoms to form the true active *OO–Ir–O₃ structure. After *OO–Ir–O₃ is desorbed, water molecules rapidly fill oxygen vacancies and are rapidly deprotonated to maintain the structural integrity of the TS–Ir/MnO₂ catalyst. These results suggest that compared with the conventional LOM mechanism, the L-LOM mechanism could accelerate the deprotonation of *OH on the surface oxygen vacancies and prevent the local peroxidation of surface Ir sites as well as the further loss of lattice oxygen atoms in the inner bulk of the catalysts. Above all, tensile strain introduced in TS–Ir/MnO₂ can enhance the covalency of the Ir–O bond to improve the deprotonation ability and increase the Ir oxidation state, triggering a continuous localized L-LOM mechanism under working conditions. Therefore, the L-LOM route can achieve much faster acidic–OER kinetics while optimizing the structural stability of the catalyst during the reaction, which endows the TS–Ir/MnO₂ catalyst with good potential for industrial applications.

**PEMWE device performance**

To verify the potential of TS-Ir/MnO₂ for industrial applications, we constructed a PEMWE device using a Nafion™ 117 membrane and TS-Ir/MnO₂ and commercial Pt/Ti as anode and cathode catalysts, respectively, in an acidic electrolyte. Figure 6a and b show pictures of both a PEM electrolyzer cell and system. The anode TS-Ir/MnO₂ and cathode Pt/Ti constitute the electrode, and the heating device and solution circulation system are matched to form the entire electrolytic cell device, in which the operating temperature is controlled at 80 °C and the solution circulation speed is 50 mL/min. The resultant electrolyzer fabricated using both TS-Ir/MnO₂ and Pt/Ti delivers improved water electrolysis activity and stability. Specifically, when the precious-metal (Ir) loading is 39 μg cm⁻² for the resultant electrolyzer with TS–Ir/MnO₂, the dosage is reduced by 54-fold compared to the target (2 mg cm⁻²) set by IRENA. This means that the cost of catalysts for

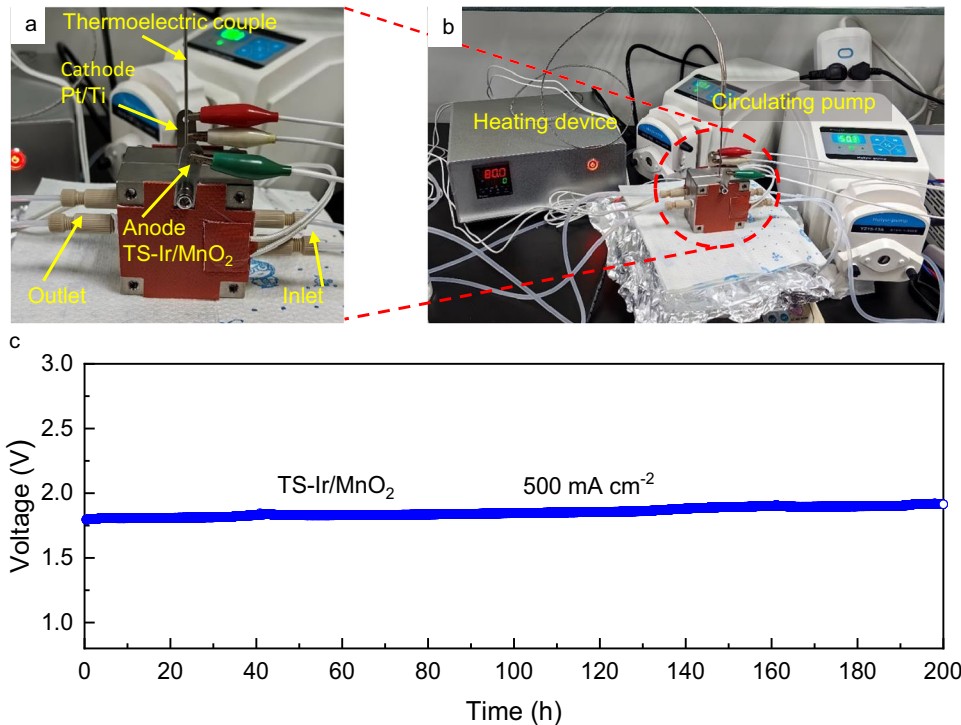

**Fig. 6 | PEMWE device performance measurements in 0.1 M HClO₄ (pH = 1).** The PEM electrolyzer (**a**) cell and system (**b**). **c** The cell voltage of the PEM electrolyzer system held at 500 mA cm⁻². The loading of catalyst is 1 mg cm⁻², and the solution resistance is 2.2 Ω.

producing hydrogen can be substantially reduced. In practical applications, catalytic stability is even more important than catalytic activity. As shown in Fig. 6c, the resultant electrolyzer delivers a current density of 500 mA cm⁻² at an applied bias of -1.75 V and can continuously operate for 200 h with negligible attenuation under simulated industrial conditions (at 80 °C). TEM images (Supplementary Fig. 32) reveal that the TS–Ir/MnO₂ morphology did not change after 200 h of operation in the PEM electrolyzer. Moreover, the ICP–OES results further confirmed that Ir negligibly dissolved (within 8%) in the samples. These stability test results indicate the good potential of the TS–Ir/MnO₂ catalyst for practical PEMWE applications.

In summary, a low-iridium electrocatalyst in which iridium atoms are localized at tensile-strained manganese-oxide surface cation sites (TS-Ir/MnO₂) was developed via cation exchange and subsequent RTAC. Mechanistic studies that employed in situ SRIR and XAFS spectroscopies revealed that the generated *OO–Ir–O₃ intermediates could be rapidly deprotonated and then water molecules rapidly filled oxygen vacancies around surface tensile–strained Ir sites to maintain the structural integrity of the catalyst, which efficiently catalyzed the OER via a continuous L-LOM mechanism. Especially, this L-LOM mechanism could considerably accelerate the acidic–OER kinetics while optimizing the material's structural integrity. Therefore, the as-obtained TS–Ir/MnO₂ catalyst delivers a high mass activity of 1025 A g$_{Ir}^{-1}$ at an overpotential of 198 mV (typically at a current density of 10 mA cm⁻²), which is approximately 19 and 380 times higher than those of Ir–MnO₂ and commercial IrO₂, respectively. Interestingly, TS–Ir/MnO₂ shows stable acidic–water electrolysis for 100 h at a current density of 200 mA cm⁻² with negligible performance degradation. More importantly, the resultant PEM electrolyzer delivered a current density of 500 mA cm⁻² at an applied bias of -1.75 V for over 200 h under simulated industrial conditions (at 80 °C). The findings in this work not only demonstrate a robust and highly-active catalyst to promote the industrial applications of PEMWEs, but also provide a strategy of using tensile straining to induce a L-LOM mechanism for enhancing the activity and durability of various catalytic reactions.

## Methods

### Synthesis of MnO₂

MnO₂ was hydrothermally synthesized using a typical procedure as follows: manganese sulfate (0.02 mol), ammonium persulfate (0.02 mol), and ammonium sulfate (0.06 mol) were dissolved in 70 mL of deionized (DI) water and ultrasonicated. Then, the solution was vigorously magnetically stirred for 1 h, transferred to an autoclave, and heated to 140 °C for 12 h. Subsequently, a black MnO₂ powder was obtained after centrifugal washing and subsequent vacuum drying several times each.

### Synthesis of TS–Ir/MnO₂

Typically, 120 mg of MnO₂ was dissolved in 30 mL of deionized water and continuously ultrasonicated for 30 min. Then, 10 mL of aqueous iridium chloride (5 mM) was added to this solution and continuously ultrasonicated. Next, this solution was continuously magnetically stirred for 12 h and then washed three times. Finally, a dried powder was obtained and transferred directly to a muffle furnace at 250 °C for 2 h and then quickly transferred to the air for rapid cooling. Lattice tensile strain was introduced to the TS–Ir/MnO₂ sample via extreme heating and cooling.

### Synthesis of Ir–MnO₂

Although this synthesis path is similar to that for TS–Ir/MnO₂, rapid heating and cooling were not required. Instead, the dried powder was transferred to the muffle furnace and heated at 5 °C min⁻¹ to 250 °C, held there for 2 h, and cooled to room temperature in the furnace.

### Structural characterizations

Spherical aberration-corrected high-angle annular dark-field scanning TEM (HAADF-STEM) and energy-dispersive spectroscopy were conducted using an FEI Titan Themis (300 kV) microscope. TEM and high-resolution TEM (HRTEM) were performed using a JEM-2100F microscope operating at 200 kV acceleration. Field-emission scanning electron microscopy images were acquired using a Gemini SEM

500 scanning electron microscope. Powder X-ray diffraction patterns were measured using a Philips X'Pert Pro Super X-ray diffractometer equipped with a Cu Kα radiation source ($\lambda = 1.54178$ Å). XPS was recorded using a Thermo ESCALAB 250Xi equipped with an Al Kα ($hv = 1486.6$ eV) excitation source. The binding energies obtained from XPS spectra were corrected with respect to the C 1$s$ peak at 284.5 eV. Inductively coupled plasma atomic emission spectroscopy (ICP−OES) was performed using an Optima 7300 DV instrument (Perkin-Elmer).

## Electrochemical measurements

All the electrochemical measurements for the acidic−OER performance were performed using a three-electrode system and the CHI760E workstation. The electrolyte was 0.1 M HClO$_4$, and graphite rods, an Ag/AgCl electrode, and a $1 \times 1$ cm$^2$ carbon cloth (CC) supported the catalysts as the counter, reference, and working electrodes, respectively. Catalyst solutions were prepared by mixing 5.0 mg of the catalyst in a solution containing 250 μL of ethanol, 730 μL of DI water, and 20 μL of a 5 wt% Nafion™ solution and sonicating the mixture to form homogeneous inks. Then, 50 μL of well-dispersed catalyst ink was carefully dropped onto the clean CC ($1 \times 1$ cm$^2$) and naturally dried for evaluation. All the final potentials were converted to corresponding reference potentials of a RHE with the equation $E$ (vs. RHE) = $E$ (vs. Ag/AgCl) + 0.197 V + 0.059 × pH (1). Linear sweep polarization curves were scanned at 5 mV s$^{-1}$ and compensated using iR correction. The electrochemical double-layer capacitance ($C_{dl}$) was calculated based on cyclic voltammetry curves that were scanned in the range 1.05-1.15 V $vs.$ RHE at 5, 10, 15, 20, 25, and 30 mV s$^{-1}$. The ECSA of the catalyst was calculated based on $C_{dl}$ according to the equation ECSA = $C_{dl}/C_s$ (2), where $C_s$ is 0.035 mF cm$^{-2}$ based on typical reported values[44]. The roughness factor (RF) was calculated by estimating the ECSA and dividing it by the geometric area of the electrode according to the equation RF = ECSA/$S_{geo}$ (3). The specific activity refers to the specific current density per ECSA ($J_{ECSA}$) and can be calculated using the equation $J_{ECSA} = J_{geo}/RF$ (4)[4].

## Mass activity and turnover frequency calculations

The mass activity (MA) (A g$^{-1}$) can be calculated based on the Ir metal loading ($m$) and current density ($j$) at typical overpotentials of 198 and 260 mV, respectively, as follows:

$$MA = \frac{j}{m} \quad (5)$$

The TOF can be calculated based on the number of active sites and current density ($j$) as follows:

$$TOF = \frac{j * N_A * 10^3}{4 * F * \text{surface} - \text{active sites(mol)}} \quad (6)$$

where $F$ is the Faraday constant (96,485 c cm$^{-1}$), and $N_A$ represents the Avogadro constant ($6.02 \times 10^{23}$). For the electrocatalysts, the total number of Ir surface sites are regarded as surface-active sites because Ir is atomically dispersed and anchored to thin MnO$_2$ nanofibers[45].

## S-number calculations

The S-number is the ratio between the amounts of evolved oxygen (calculated from $Q$ total) and dissolved iridium (obtained from ICP−MS measurements)[33].

## In situ SRIR measurements

In situ SRIR spectra were measured using homemade cells at the infrared beamline (BL01B) of the National Synchrotron Radiation Laboratory (China). An infrared (IR) spectrometer (Bruker 66 V s$^{-1}$) equipped with a KBr beam splitter, a liquid nitrogen-cooled MCT detector, and an IR microscope (Bruker Hyperion 3000) that had a 16×

objective lens comprised the SRIR testing device. SRIR spectroscopy can provide a clear spectrum in the range 15−4000 cm$^{-1}$ at a high spectral resolution of 0.25 cm$^{-1}$. To reduce the interference of water molecules on the infrared signal during the electrochemical reaction, the distance between the electrode and window must be adjusted at the microscale. To obtain a sufficiently intense reaction-intermediate signal, the reflection mode must be used to collect infrared signals. The background spectrum was acquired for the electrocatalyst electrode at the open-circuit voltage before each systemic OER measurement, and the measured OER potential ranges were 1.0−1.55 V. Notably, in situ SRIR curves were smoothed using data processing for 10 points, which means that 10 consecutive points were averaged to improve the signal-to-noise ratio of the curves.

## In situ XAFS measurements

In situ XAFS spectra were measured using homemade cells at both the 1W1B station at the Beijing Synchrotron Radiation Facility (BSRF) and BL14W1 at the Shanghai Synchrotron Radiation Facility (SSRF) in China. The BSRF and SSRF storage rings were operated at 2.5 and 3.5 GeV, respectively, at a maximum current of 250 mA. The beam originating from the bending magnet was monochromatized utilizing a Si (111) double-crystal monochromator and further detuned by 15% to remove any higher harmonics. XAFS spectra were collected using a 19-element solid-state detector operating in fluorescence mode. In situ XAFS spectra were measured using catalyst-modified CC as the working electrode that was taped with Kapton® film on the back. To obtain information about the active site evolution during the electrochemical reaction, a series of representative potentials (1.15−1.43 V) were applied to the electrode. During the collection of XAFS measurements, the absorption edge ($E_0$) position was calibrated using a standard Ir sample.

## XAFS data analysis

The obtained EXAFS data were processed using the ATHENA module that was implemented in IFEFFIT software packages[46]. Subsequently, to separate EXAFS contributions from different coordination shells, the EXAFS data were Fourier-transformed to real space using a Hanning window (d$k = 1.0$ Å$^{-1}$) in the $k$-space range 2.6−10.8 Å$^{-1}$. The amplitude reduction factor ($S_0^2 = 0.78$) was obtained from the fitted $L_3$-edge EXAFS curve for iridium dioxide. The curve was fitted using the $k^2$-weighted EXAFS function χ($k$) data in $k$- and $R$-ranges 2.6−10.8 Å$^{-1}$ and 1.0−2.4 Å, respectively. The number of independent points can be calculated using the equation $N_{ipt} = 2\Delta k \Delta R/\pi = 2 \times (10.8 - 2.6) \times (2.4 - 1.0)/\pi = 7$ (7). For the ex situ condition of TS−Ir/MnO$_2$, the FT curve showed a prominent coordination peak at 1.54 Å, which was assigned to the Ir−O coordination bond. Compared with the ex situ condition, the first-shell coordination peaks at applied biases of 1.15, 1.30, and 1.43 V were stronger and had higher R shifts, which were ascribed to oxygen species adsorption. Therefore, the addition of Ir−O coordination bonds was considered during curve fitting. Notably, the peak at 1.43 V, which corresponded to Ir−O coordination bonds, weakened, which may be attributed to the evolution of the Ir local coordination structure. During curve fitting, the Debye−Waller factors ($\sigma^2$), coordination numbers ($N$), interatomic distances ($R$), and energy shifts ($\Delta E_0$) were treated as adjustable parameters. To reduce the number of adjustable fitting parameters, $\Delta E_0$ was the same as that of the ex situ sample at different applied potentials.

## Electrochemical measurements in PEMWE tests

First, the as-obtained Nafion™ 117 membrane had be pretreated by cleaning with a solution comprising 3 wt% H$_2$O$_2$, 0.5 M H$_2$SO$_4$, and deionized water at 80 °C for 1 h and was preserved in distilled water for subsequent PEMWE evaluations. Then, TS−Ir/MnO$_2$ and Pt/Ti (Pt-plated Ti-fiber felt) were used as anodic and cathodic catalysts for the PEMWE tests, respectively. To prepare the anodic catalyst ink, 5 mg of

the catalyst was suspended in 1 mL of ethanol and water mixed in a 1:3 volumetric ratio. Then, 30 μL of the 5 wt% Nafion™ solution was added to this mixture, and the suspension was ultrasonicated for 1 h until a well-dispersed catalyst ink was formed. The as-prepared TS–Ir/MnO₂ catalyst ink (200 μL) was sprayed on sheets of Ti-fiber felt (surface area $1 \times 1$ cm²). Then, the electrode, which comprised the Ti-fiber felt-supported TS–Ir/MnO₂, treated Nafion™ 117 membrane, and Pt/Ti, was hot pressed at a certain pressure. Finally, the electrode was applied in PEMWE for evaluating catalytic performance. The PEMWE device was operated at 80 °C using a 0.1 M HClO₄ electrolyte flowing at 50 mL min⁻¹. The stability of the PEMWE conducted using the TS–Ir/MnO₂ anodic catalyst was evaluated by measuring chronopotentiometry at a current density of 500 mA cm⁻² for 200 h at 80 °C and ambient pressure. All the cell voltages measured in PEMWE electrolyzers were reported without applying any iR compensation.

## Data availability

All data reported in this paper are available from the corresponding author upon request.

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

## Acknowledgements

This work was supported by the National Key R&D Program of China (2022YFA1502903 (Q.L.)), the National Natural Science Foundation of China (12205300 (H.S.), 22241202 (Q.L.), and 12135012 (H.S.)), and the Natural Science Foundation of Anhui Province (Grant Nos. 2208085J01 (Q.L.) and 2208085QA28 (H.S.)).

## Author contributions
Q.L., H.S., and S.L. conceived the project. H.S. and C.Y. designed the in situ SRIR and XAFS experiments. H.S. and C.Y. conducted the experiments, including catalyst synthesis, catalytic tests, and in situ SRIR and XAFS measurements. M.L. performed catalytic activity. X.Z. and K.Z. performed TEM tests. W.Z., Y.Z., and S.L. analyzed the experimental data. The manuscript was written by H.S., C.Y., and Q.L. All authors discussed the results and commented on the paper.

## Competing interests
The authors declare no competing interests.
