## [Peer review file · Nature Communications]

REVIEWER COMMENTS

Reviewer #1 (Remarks to the Author):

The contribution from Su et al. is a very good contribution. The noteworthy and important result is to show that an earth-abundant metal oxide (MnO_2) can serve as a host for the precious metal Ir (low loading) for application at low pH. The contribution, however, is not entirely novel since the addition of Ir to MnO_2 is not new, although the authors develop a synthetic protocol that improves activity and importantly the stability of the metal oxide at low pH. What I like most about the contribution, however, is the mechanistic study the authors present. They show results that support their contention that a strained Ir-O-Mn structure allows the lattice oxygen mechanism to operate and produce O_2 , but the stability of the catalyst is maintained due to the replacement of the O by water.

In its present form, I would not recommend publication. A major revision is needed. I have several comments/suggestions:

1) The authors are light on reporting on prior work in this area. They do not even reference a related paper that looks at a very similar system, Ir on $\alpha\text{-MnO}_2$ (ChemElectroChem, vol. 8, p. 418, 2021). This is a significant omission. The authors need to show that their work is different. There is also another recently published papers that shows a strained $\beta\text{-MnO}_2$ lattice when Ir is incorporated. The authors need to do a better job of presenting what was done before their work.

2) The authors should include a table that presents all the EXAFS data. For example, the authors quote bond distances in the TS-materials, but a table would be a good way for the reader to compare the distances associated with the other control materials.

3) The authors state in their contribution, "Clearly, the Ir-O bond lengths are different in TS-Ir/ MnO_2 , Ir- MnO_2 , and Ir O_2 ." First, I would refrain from using "clearly". After looking at the Fig. 2 and the supporting information, I do not see convincing evidence of a change in the Ir-O bond distance in the the different samples. The authors should emphasize what the reader should be looking at. Also, they should tabulate all the EXAFS fitting parameters used in their fitting to determine the bond distances.

4) Related to the above, it is stated that "Additionally, the O 1s XPS spectra support the existence of Ir-O bonds in the sample" in Figure 9 in the supporting information. All the O 1s spectra associated with the Ir/ MnO_2 samples look similar to MnO_2 . Again, the authors need to better expand on how they are interpreting the data.

5) A comparison of the catalytic kinetic parameters of the TS-material to published results that went before for Ir/ MnO_2 would be useful.

Reviewer #2 (Remarks to the Author):

In this work, the authors suggest a localized lattice oxygen-mediated mechanism under working conditions for Ir-doped MnO_2 in acidic electrolyte, which provides an intriguing understand about OER reaction kinetics. So far, some similar catalysts (Ir- MnO_2) about OER have been reported. Therefore, some fundamental issues need to be reconsidered.

(1) A good summary about similar catalysts and relative discussions can be added to better understand your advantages.

(2) Why the rapid annealing-cooling treatment can lead to a tensile-strain in the TS-Ir MnO_2 , but traditional annealing cannot occur? What is the intrinsic reason?

(3) How to ensure that the Ir atoms are localized at surface Mn sites? Is it displacement doping or atom anchored configuration. This point is very important to understand your OER mechanism.

- (4) If the Ir atoms are doped onto MnO₂ surface, why a larger XRD shift occurs at TS-Ir/MnO₂?
- (5) Based on the XAS result, the bond length of Mn-O is hardly changed. This means that the host MnO₂ cannot give rise to a lattice strain. Based on this consideration, a small number of Ir atoms at catalyst surface really can lead to a large XRD difference? It is hard to understand this point.
- (6) In fact, any doping strategy all can lead to a change in localized region, which makes it difficult toward to the tensile strain.
- (7) In Fig. 2, the Ir-O bond in TS-Ir/MnO₂ is stretched comparing to Ir-MnO₂, the relative Mn-O bond should be compressed, especially for neighboring Mn atoms around the doped Ir sites. Why you can observe a contrary behavior? How to understand this reason?
- (8) Why the L-LOM cannot occur at Ir-MnO₂? How to understand the contribution of the localized strain onto regulating OER pathway? Some relative calculations can better support your assertions.
- (9) Why the I-t curves in Fig. 3f is slowly decreased? But the V-t curves in Fig. 6c are slowly increased.
- (10) How to estimate the number of the active sites in different catalyst? In other words, how to ensure the equivalent Ir atom at MnO₂ surface between Ir-MnO₂ and TS-Ir/MnO₂.
- (11) In your OER test, some Ir atoms are partial dissociated, which may generate a dynamic balance to affect the catalytic reactivity (Nature Catalysis, 4, 1012-1023, 2021). How to exclude this factor?

Reviewer #3 (Remarks to the Author):

The manuscript by Su et al. designed a low-iridium electrocatalyst in which tensile-strained iridium atoms were introduced and localized at the manganese-oxide surface cation sites (TS-Ir/MnO₂) for efficient and stable acidic-oxygen evolution reaction (OER) activity. The choice of methods is appropriate, and the authors performed controlled experiments such as in situ X-ray absorption fine structure (XAFS) and synchrotron radiation infrared (SR-IR) spectroscopy at the OER working conditions for investigating the dynamic catalytic process and revealing the catalytic reaction mechanism of the catalyst. What impressed me the most was the authors' observation of a new localized lattice oxygen-mediated (L-LOM) mechanism in the well-designed TS-Ir/MnO₂ catalyst that can regulate the kinetics of acidic-OER and optimize the structural integrity of the catalyst during the reaction process, which is likely to be of wide interest to the catalytic mechanism understanding and catalyst design for the new energy technologies such as the proton-exchange membrane water electrolysis. Overall, I think that the whole work is of novelty and the findings are of particular significance. Therefore, I recommend the publication of this manuscript in the high-profile journal of Nature Communications after the following issues are clarified:

1. In the experimental and catalyst preparation sections, the authors claimed that tensile strain was introduced by a rapid thermal annealing-cooling strategy. As is known, the temperature of annealing will affect the structure of samples and then affect the performance of the samples. I wonder if the authors tried other temperatures, such as the lower or higher temperatures than that of used by the authors in the manuscript?
2. In the main body of the text, the authors mentioned that the localized lattice oxygen-mediated (L-LOM) mechanism was realized for TS-Ir/MnO₂. To consolidate this conclusion, the authors should strengthen the relationship between the structure (such as the atomic and/or electronic structure of the catalyst) and reaction mechanism.
3. In the electrocatalytic performance characterizations, it is known that the environment of electrochemical testing has a certain effect on the performance of the catalyst. In this regard, I suggest that the potential temperature effects should be considered and some related discussions on this issue should be included.
4. The stability of catalysts is more important than the catalytic activity for the practical applications.

A suitable explanation for the observed stability of the catalyst would open a great avenue for further developments of acidic-OER catalysts. Therefore, I strongly suggest the authors strengthening related discussions on this issue.

5. General comment. Some of the potentials listed throughout the manuscript need to be listed/provided relative to a reference electrode.

6. There are some grammatical and handwriting errors, in the page of 7 "which reveals the electron transfer from Mn to Ir.", "the main peaks in the range 1.5-1.6 Å" and "for which the corresponding data are listed in Table 1". In the page of 16 "...TS-Ir/MnO₂ and commercial Pt/C as anode and cathode catalysts". In Supplementary Information, Fig. 12 "The corresponding overpotentials at current density of 10, 100 and 200 mA cm⁻²", Fig. 13 "OER polarization curves of TS-Ir/MnO₂ of different metal loading of Ir" and Fig. 21 "(b) O 1s XPS spectra for TS-Ir/MnO₂ after electrochemical measurements". The authors should check them out carefully.

Response to Reviewers' Comments

We are grateful to the reviewers for having given us important and valuable comments on the manuscript NCOMMS-23-39607 entitled: “*Tensile straining of iridium sites in manganese oxides for proton-exchange membrane water electrolyzers*”. The detailed replies to your comments are presented in a point-to-point manner as follows. **The modifications in the revised manuscript are highlighted in the yellow background.**

Reply to Reviewer #1

The contribution from Su et al. is a very good contribution. The noteworthy and important result is to show that an earth-abundant metal oxide (MnO₂) can serve as a host for the precious metal Ir (low loading) for application at low pH. The contribution, however, is not entirely novel since the addition of Ir to MnO₂ is not new, although the authors develop a synthetic protocol that improves activity and importantly the stability of the metal oxide at low pH. What I like most about the contribution, however, is the mechanistic study the authors present. They show results that support their contention that a strained Ir-O-Mn structure allows the lattice oxygen mechanism to operate and produce O₂, but the stability of the catalyst is maintained due to the replacement of the O by water.

In its present form, I would not recommend publication. A major revision is needed. I have several comments/suggestions:

We greatly appreciate Reviewer #1 for highlighting the major findings of our work. After reading your comments and suggestions, we have seriously addressed these questions and revised our manuscript carefully. We hope the revised manuscript could meet the high standards of *Nature Communications*.

1. Comment: The authors are light on reporting on prior work in this area. They do not even reference a related paper that looks at a very similar system, Ir on alpha-MnO₂ (Chem Electro Chem, vol. 8, p. 418, 2021). This is a significant omission. The authors need to show that their work is different. There is also another recently published papers

that shows a strained beta-MnO₂ lattice when Ir is incorporated. The authors need to do a better job of presenting what was done before their work.

Reply: Thank you for your insightful comment and constructive suggestion that is quite useful for improving the quality of this work. According to your suggestion, relevant prior work and innovations are summarized and analyzed in the revised manuscript.

MnO₂ is an abundant and inexpensive transition metal oxide with high surface activity and catalytic stability in acidic media, which is considered as a potential carrier for acidic OER reactions. The α -MnO₂ supported IrO₂ catalyst also showed potential for acidic oxygen evolution due to its high active specific surface areas, high active-site densities and low charge transfer resistance in a one-dimensional α -MnO₂ structure (J. Phys. Chem. C 2008, 112, 4406-4417). To increase activity and reduce the loading of precious metals, a low loading of 5% iridium supported on α -MnO₂ electrocatalyst exhibited excellent electrocatalytic OER performance, which can be attributed to smaller iridium particles and more high-valence-state iridium (Chem. Electro. Chem 2021, 8, 418-424). Recently, Lee's group reported Ru-atom-array patches supported on α -MnO₂ for an efficient OER process following an oxide path mechanism (Nat. Catal. 2021, 4, 1012-1023). Furthermore, the high-loading 22 wt% Ir-incorporated β -MnO₂ exhibited enhanced OER activity and stability, attributing to a strained lattice containing optimized Ir–O–Ir and Ir–O–Mn bonding along with the presence of Mn³⁺ (Chem. Cat. Chem. 2023, 15, e202201549). However, the formation of *OOH over metal sites is the rate-limiting step, which is considered as the bottleneck in improving the acidic-OER performance. Moreover, slow deprotonation at lattice oxygen-atom sites and a large number of oxygen vacancies can lead to rapid degradation of performance. Therefore, simultaneously breaking the linear relationship of multiple reaction intermediates and inhibiting a large number of oxygen vacancies is essentially desirable for designing advanced electrocatalysts.

In this manuscript, a low-iridium electrocatalyst in which tensile-strained iridium atoms are localized at manganese-oxide surface cation sites (TS-Ir/MnO₂) was successfully synthesized, where the strong interaction between metal atoms and support

optimizes the covalency of Ir–O bonds that can profoundly determine the catalytic OER performance and reaction mechanism. First, the presence of tensile strain can efficiently regulate the covalency of Ir–O bonds in TS-Ir/MnO₂ to realize a high oxidation state of Ir sites, which can accelerate the reaction kinetics. Second, *in situ* synchrotron characterizations reveal that the well-designed TS-Ir/MnO₂ electrocatalyst can trigger a new continuous localized lattice oxygen-mediated (L-LOM) mechanism under working conditions, successfully breaking the linear relationship of multiple reaction intermediates for efficient OER activity. Particularly, the L-LOM process could substantially boost the adsorption and transformation of H₂O molecules over the oxygen vacancies around the tensile-strained Ir sites and prevent further loss of lattice oxygen atoms in the inner MnO₂ bulk to optimize the structural integrity of the catalyst for long-term OER stability. Finally, TS-Ir/MnO₂ delivers a current density of 500 mA cm⁻² and operates stably for 200 h with no significant performance degradation, suggesting the excellent potential of TS-Ir/MnO₂ for practical application in PEMWE devices.

Accordingly, in line 7, page 3 of the revised manuscript, the following text **has been modified**: “Recently, MnO₂ is an abundant and inexpensive transition metal oxide with high surface activity and catalytic stability in acidic media, which is considered as a potential carrier for acidic OER reactions. The IrO₂ nanoparticles and a low loading of 5% iridium supported on the α -MnO₂ electrocatalyst exhibited excellent electrocatalytic OER performance due to high active specific surface areas and more high-valence-state iridium (*J. Phys. Chem. C* 2008, 112, 4406-4417, *Chem. Electro. Chem* 2021, 8, 418-424). To break the linear relationship of multiple reaction intermediates for increasing activity, Ru-atom-array patches supported on α -MnO₂ with appropriate atomic distances in symmetric dual-metal sites delivered enhanced acidic-OER activity following an oxide path mechanism (*Nat. Catal.* 2021, 4, 1012-1023). Furthermore, a strained lattice containing optimized Ir–O–Ir and Ir–O–Mn bonding along with the presence of Mn³⁺ in the high-loading 22 wt% Ir-incorporated β -MnO₂ exhibited enhanced OER stability (*Chem. Cat. Chem.* 2023, 15, e202201549).

Therefore, simultaneously breaking the linear relationship of multiple reaction intermediates and inhibiting a large number of oxygen vacancies is essentially desirable for designing advanced electrocatalysts, but it is a great challenge.”

2. Comment: The authors should include a table that presents all the EXAFS data. For example, the authors quote bond distances in the TS-materials, but a table would be a good way for the reader to compare the distances associated with the other control materials.

Reply: This is a very good question. According to your suggestion, the apparent bond lengths of all samples according to the EXAFS data are summarized in Table N1. For all samples under air conditions, the Ir–O bond length in TS-Ir/MnO₂ (1.54 Å) is shorter than that in IrO₂ (1.60 Å) and longer than that in Ir-MnO₂ (1.48 Å), suggesting that suitable covalency and electron transfer of the Ir–O bond were realized. During the OER process, the first-shell peak exhibited a slight high-R shift from 1.54 to 1.59 Å with the condition changing from the *ex situ* state to an applied potential of 1.15 V (Table N2), which implies the rearrangement of the local coordination structure of Ir sites during the OER due to the adsorption of oxygen species over Ir sites.

Table N1. The apparent bond length of Ir–O/Mn–O in all samples.

Sample	Apparent bond length (Ir–O)	Apparent bond length (Mn–O)
TS-Ir/MnO ₂	1.54 Å	1.50 Å
Ir-MnO ₂	1.48 Å	1.47 Å
MnO ₂	--	1.47 Å
IrO ₂	1.60 Å	--

Table N2. The apparent bond length of Ir–O in TS-Ir/MnO₂ under different working conditions.

Sample	Apparent bond length (Ir–O)
ex situ	1.54 Å
1.15 V	1.59 Å
1.30 V	1.59 Å
1.45 V	1.59 Å

Accordingly, Tables N1 and N2 have been added in the Supplementary Information as Supplementary Tables 1 and 6. In reversed line 11, page 8 of the revised manuscript, the following text **has been modified**: “Fig. 2d, Supplementary Fig. 10 and Supplementary Table 1 show that the dominant coordination peak at 1.48 Å, which is assigned to the first shell of Ir–O in Ir–MnO₂, is shorter than that at 1.60 Å (for Ir–O bonds in IrO₂), which is attributed to the distinct second shell structure between Ir–MnO₂ and IrO₂.” In line 1, page 16 of the revised manuscript, the following text **has been modified**: “Compared with the ex situ state, the first-shell peak intensified at an applied potential of 1.15 V and exhibited a slight high-R shift from 1.54 to 1.59 Å (Supplementary Table 6). With increasing applied potential to 1.30 V, the Ir–O coordination peak further intensified.”

3. Comment: The authors state in their contribution, "Clearly, the Ir–O bond lengths are different in TS-Ir/MnO₂, Ir–MnO₂, and IrO₂." First, I would refrain from using "clearly". After looking at the Fig. 2 and the supporting information, I do not see convincing evidence of a change in the Ir–O bond distance in the the different samples. The authors should emphasize what the reader should be looking at. Also, they should tabulate all the EXAFS fitting parameters used in their fitting to determine the bond distances.

Reply: Thank you for your nice questions. We are sorry that the changes in the Ir–O bond distance were not clearly addressed in the previous manuscript. According to your suggestion, the real lengths of Ir–O bonds were fitted according to EXAFS data and are

summarized in Table N2. The Ir–O real bond length in TS-Ir/MnO₂ (1.94 Å) is shorter than that in IrO₂ (2.01 Å) and longer than that in Ir-MnO₂ (1.89 Å).

Specifically, as shown in Fig. 2d, the apparent bond length of the Ir-O coordination can be observed from FT curves of the Ir *L*₃-edge EXAFS spectra. The main peaks in the range of 1.45-1.60 Å can be assigned to the first-shell of Ir–O coordination. For Ir-MnO₂, the apparent bond length of Ir–O is 1.48 Å, which is slightly shorter than that of IrO₂ (1.60 Å), which is attributed to the distinct second shell structure between Ir–MnO₂ and IrO₂. After tensile strain is introduced in TS-Ir/MnO₂, the apparent bond length of Ir–O increases to 1.54 Å. In order to indeed evaluate the real bond length of Ir–O coordination, the fitted curves of the *k*²-weighted Ir *L*₃-edge EXAFS spectra of all samples in Figure N1 and the corresponding fitting results in Table N2 show that the real Ir–O bond length is 1.94 Å in TS-Ir/MnO₂, which is shorter than that in IrO₂ (2.01 Å) and longer than that in Ir-MnO₂ (1.89 Å), suggesting the existence of tensile strain and increased covalency.

Figure N1. The fitting curves of k^2 -weighted Ir L_3 -edge EXAFS spectra and the $Re(k^2\chi(k))$ oscillation curves for TS-Ir/MnO₂ (a and b), Ir-MnO₂ (c and d) and IrO₂ (e and f).

Table N3. Structural parameters for the TS-Ir/MnO₂, Ir-MnO₂ and IrO₂ electrocatalysts extracted from quantitative EXAFS curve fitting using the ARTEMIS module of IFEFFIT.

Sample	Path	N	R (Å)	$\sigma^2(10^{-3}\text{Å}^2)$	$\Delta E_0(\text{eV})$	R-factor
TS-Ir/MnO ₂	Ir-O	4.2±0.3	1.94±0.005	3.6±0.5	9.7±1.3	0.005
Ir-MnO ₂	Ir-O	4.1±0.2	1.89±0.01	5.1±1.1	9.8±2.1	0.005
IrO ₂	Ir-O	6	2.01±0.01	3.3±0.5	9.7±1.6	0.003

Accordingly, in reversed line 13, page 8 of the revised manuscript, “Clearly, the Ir–O bond lengths are different in TS-Ir/MnO₂, Ir–MnO₂, and IrO₂.” has been changed to “A slight difference in the position of the main peak at approximately 1.5 Å in Fig. 2d

represents a different apparent Ir–O bond length in TS-Ir/MnO₂, Ir-MnO₂, and IrO₂.” Furthermore, Supplementary Fig. 11 and Supplementary Table 2 have been replaced by Figure N1 and Table N3 in the Supplementary Information. In reversed line 3, page 8 of the revised manuscript, the following text **has been added**: *“In order to indeed evaluate the real bond length of the Ir–O coordination, the fitted curves of the k²-weighted Ir L₃-edge EXAFS spectra of all samples in Supplementary Fig. 11, for which the corresponding data are listed in Supplementary Table 2, show that the coordination number of the Ir–O first shell is ~4 and that the real Ir–O bond length is 1.94 Å in TS-Ir/MnO₂, which is shorter than that in IrO₂ (2.01 Å) and longer than that in Ir-MnO₂ (1.89 Å), suggesting the existence of tensile strain and increased covalency.”*

4. Comment: Related to the above, it is stated that "Additionally, the O 1s XPS spectra support the existence of Ir–O bonds in the sample" in Figure 9 in the supporting information. All the O 1s spectra associated with the Ir/MnO₂ samples look similar to MnO₂. Again, the authors need to better expand on how they are interpreting the data.

Reply: Thank you for your careful consideration, and it is useful for improving the quality of this work. The discussions about the XPS spectra are stated as follows.

XPS is an important characterization method to obtain the composition and valence state of samples, and its higher binding energy represents the higher valence state of the element. As shown in Figure 2a, the Ir 4f XPS spectrum shows a higher oxidation state of Ir after introducing Ir atoms and tensile strain in comparison with IrO₂. Correspondingly, in order to further clarify the composition of lattice O, we re-fit the lattice O, as shown in Figure N2. The only fitting peak of 529.76 eV in the lattice O spectrum of MnO₂ can be assigned to Mn–O. For comparison, the two peaks at 529.70 and 529.91 eV are Mn–O and Ir–O, respectively, suggesting the electron interaction between Ir and Mn species. After introducing the tensile strain in TS-Ir/MnO₂, the peak of Ir–O (530.18 eV) moves toward high energy, suggesting the increased oxidation state of Ir due to electron transfer driven by tensile strain.

Figure N2. High-resolution spectra of O 1s for MnO₂ (a), Ir-MnO₂ (b) and TS-Ir/MnO₂ (c).

Accordingly, Supplementary Fig. 9 has been replaced by Fig. N2 in the Supplementary Information. In line 15, page 8 of the revised manuscript, the following text **has been added**: “Furthermore, the higher binding energy of Ir–O in TS-Ir/MnO₂ further demonstrates that the increased oxidation state of Ir due to electron transfer driven by tensile strain.”

5. Comment: A comparison of the catalytic kinetic parameters of the TS-material to published results that went before for Ir/MnO₂ would be useful.

Reply: Thank you for your constructive suggestions that are quite useful for improving the integrity of this work. According to your suggestion, a comparison of the catalytic kinetic parameters of TS-Ir/MnO₂ with those of state-of-the-art electrocatalysts (Table N4) is provided. It can be seen that the TS-Ir/MnO₂ is one of the comparable OER performance reported electrocatalysts in terms of catalytic kinetic parameters such as the Tafel slope.

The Tafel slope is an important catalytic kinetic parameter for the evaluation of the activity of electrocatalysts. As seen from Table N4, the Tafel slope of TS-Ir/MnO₂ is 56.6 mV dec⁻¹, which is smaller than that of the reported catalysts supported on carbon paper or carbon cloth (Angew. Chem. Int. Ed. 2023, 62, e202301128). Based on the comparison results in Table N4, TS-Ir/MnO₂ delivers a low Tafel slope, which is beyond that of comparable OER catalysts.

Accordingly, Table N4 has been added in the Supplementary Information as Supplementary Table 3, and in line 3 of page 11 of the revised manuscript, the following text **has been added**: “TS-Ir/MnO₂ delivers an efficient catalytic kinetic activity in Supplementary Table 3, which is beyond that of comparable OER catalysts.”

Table N4. Comparison of the OER activity of the TS-Ir/MnO₂ electrocatalyst with other recently reported catalysts in acidic solution according to the catalytic kinetic parameter (Tafel slope).

Catalyst	Electrolyte	Support	Tafel slope (mV dec ⁻¹)	Ref.
TS-Ir/MnO ₂	0.1 M HClO ₄	Carbon cloth	56.6	This work
Ir-MnO ₂	0.1 M HClO ₄	Carbon cloth	101.2	This work
IrMnOF@Ir	0.1 M HClO ₄	Carbon paper	58.3	Angew.Chem. Int. Ed. 2023, 62, e202301128
Ir/δ-MnO ₂	0.5 M H ₂ SO ₄	Carbon Paper	123	ChemCatChem 2023, 15, e202201549
(Mn _{0.8} Ir _{0.2})O ₂ ·10F	0.5 M H ₂ SO ₄	Ti-foils	38	ACS Catal. 2019, 9, 2134-2157
IrO _x /Zr ₂ ON ₂	0.5 M H ₂ SO ₄	RDE	48	Adv. Funct. Mater. 2023. 33, 2301557
GB-Ta _{0.1} Tm _{0.1} Ir _{0.8} O _{2-δ}	0.5 M H ₂ SO ₄	RDE (Au)	64	Nat. Nanotechno. 2021, 16, 1371-1377.
IrO ₂ /GCN	0.5 M H ₂ SO ₄	RDE	57	Angew.Chem. Int. Ed. 2019, 58, 12540-12544
12Ru/MnO ₂	0.1 M HClO ₄	GCE	29.4	Nat. Catal. 2021, 4, 1012-1023
H/d-MnO _x /RuO ₂	0.5 M H ₂ SO ₄	GCE	43.8	Adv. Funct. Mater. 2023, 2307010
Li _{0.52} RuO ₂	0.5 M H ₂ SO ₄	GCE	83.3	Nat. Commun. 2022, 13, 3784.
90-Co-MnO ₂	0.1 M HClO ₄	GCE	158	Adv. Mater. 2023, 35, 2207066
IrO ₂ /a-MnO ₂	0.1 M HClO ₄	GCE	74	ChemElectroChem 2021, 8, 418–424

Reply to Reviewer #2

In this work, the authors suggest a localized lattice oxygen-mediated mechanism under working conditions for Ir-doped MnO₂ in acidic electrolyte, which provides an intriguing understand about OER reaction kinetics. So far, some similar catalysts (Ir-MnO₂) about OER have been reported. Therefore, some fundamental issues need to be reconsidered.

We are very grateful to Reviewer #2 for highlighting the major findings of the

catalytic mechanism. According to your constructive comments and suggestion, we have carefully addressed these questions and revised our manuscript.

1. Comment: A good summary about similar catalysts and relative discussions can be added to better understand your advantages.

Reply: Thank you for your insightful comment and constructive suggestion that is quite useful for improving the quality of this work. According to your suggestion, relevant prior work and innovations are summarized and analyzed in the revised manuscript.

MnO₂ is an abundant and inexpensive transition metal oxide with high surface activity and catalytic stability in acidic media, which is considered as a potential carrier for acidic OER reactions. The α -MnO₂-supported IrO₂ catalyst also showed potential for acidic oxygen evolution due to its high active specific surface areas, high active-site densities and low charge transfer resistance in one-dimensional α -MnO₂ structure (J. Phys. Chem. C 2008, 112, 4406-4417). To increase activity and reduce the loading of precious metals, a low loading of 5% iridium supported on α -MnO₂ electrocatalyst exhibited excellent electrocatalytic OER performance, which can be attributed to smaller iridium particles and more high-valence-state iridium (Chem. Electro. Chem 2021, 8, 418-424). Recently, Lee's group reported Ru-atom-array patches supported on α -MnO₂ for an efficient OER process following an oxide path mechanism (Nat. Catal. 2021, 4, 1012-1023). Furthermore, the high-loading 22 wt% Ir-incorporated β -MnO₂ exhibited enhanced OER activity and stability, attributing to a strained lattice containing optimized Ir–O–Ir and Ir–O–Mn bonding along with the presence of Mn³⁺ (Chem. Cat. Chem. 2023, 15, e202201549). However, the formation of *OOH over metal sites is the rate-limiting step, which is considered as the bottleneck in improving the acidic-OER performance. Moreover, slow deprotonation at lattice oxygen-atom sites and a large number of oxygen vacancies can lead to rapid degradation of performance. Therefore, simultaneously breaking the linear relationship of multiple reaction intermediates and inhibiting a large number of oxygen vacancies is essentially desirable for designing advanced electrocatalysts.

In this manuscript, a low-iridium electrocatalyst in which tensile-strained iridium

atoms are localized at manganese-oxide surface cation sites (TS-Ir/MnO₂) was successfully synthesized, where the strong interaction between metal atoms and support optimizes the covalency of Ir–O bonds that can profoundly determine the catalytic OER performance and reaction mechanism. First, the presence of tensile strain can efficiently regulate the covalency of Ir–O bonds in TS-Ir/MnO₂ to realize high oxidation state of Ir sites, which can accelerate the reaction kinetics. Second, *in situ* synchrotron characterizations reveal that the well-designed TS-Ir/MnO₂ electrocatalyst can trigger a new continuous localized lattice oxygen-mediated (L-LOM) mechanism under working conditions, successfully breaking the linear relationship of multiple reaction intermediates for an efficient OER activity. Particularly, the L-LOM process could substantially boost the adsorption and transformation of H₂O molecules over the oxygen vacancies around the tensile-strained Ir sites and prevent further loss of lattice oxygen atoms in the inner MnO₂ bulk to optimize the structural integrity of the catalyst for long-term OER stability. Finally, TS-Ir/MnO₂ delivers a current density of 500 mA cm⁻² and operates stably for 200 h with no significant performance degradation, suggesting the excellent potential of TS-Ir/MnO₂ for practical application in PEMWE devices.

Accordingly, in line 7, page 2 of the revised manuscript, the following text **has been modified**: “Recently, MnO₂ is an abundant and inexpensive transition metal oxide with high surface activity and catalytic stability in acidic media, which is considered as a potential carrier for acidic OER reactions. The IrO₂ nanoparticles and a low loading of 5% iridium supported on the α -MnO₂ electrocatalyst exhibited excellent electrocatalytic OER performance due to high active specific surface areas and more high-valence-state iridium (*J. Phys. Chem. C* 2008, 112, 4406-4417, *Chem. Electro. Chem* 2021, 8, 418-424). To break the linear relationship of multiple reaction intermediates for increasing activity, Ru-atom-array patches supported on α -MnO₂ with appropriate atomic distances in symmetric dual-metal sites delivered enhanced acidic-OER activity following an oxide path mechanism (*Nat. Catal.* 2021, 4, 1012-1023). Furthermore, a strained lattice containing optimized Ir–O–Ir and Ir–O–Mn

bonding along with the presence of Mn³⁺ in the high-loading 22 wt% Ir-incorporated β -MnO₂ exhibited enhanced OER stability (Chem. Cat. Chem. 2023, 15, e202201549). Therefore, simultaneously breaking the linear relationship of multiple reaction intermediates and inhibiting a large number of oxygen vacancies is essentially desirable for designing advanced electrocatalysts, but it is a great challenge.”

2. Comment: Why the rapid annealing-cooling treatment can lead to a tensile-strain in the TS-IrMnO₂, but traditional annealing cannot occur? What is the intrinsic reason?

Reply: This is a very nice question. The catalytic activity of various types of electrochemical reactions can also be enhanced by strain engineering, usually in the form of compressive or tensile surface strains in carefully engineered core shells or alloyed nanostructures of noble metal catalysts (Science 2016, 352, 73-76; Nature 2019, 574, 81-85). Furthermore, the heat- and light-drive are considered effective strategies to introduce stress into the lattice, and then optimize the electronic structure to accelerate the reaction kinetics of water decomposition and oxygen reduction (Nat. Nanotechnol. 2021, 16, 1371-1377; Nat. Energy 2019, 4, 115-122). In this work, Ir atoms are first introduced to the MnO₂ carrier by ion exchange and then transferred directly to the Muffle furnace at 250 °C for 2 h, in which rapid heat treatment introduces stress in the lattice. After that, the hot sample is transferred directly to room temperature for rapid cooling, when the lattice stress does not undergo a slow cooling process for stress release. Therefore, the rapid annealing-cooling treatment can lead to a tensile-strain retention in TS-Ir/MnO₂. For comparison, Ir-MnO₂ undergoes a traditional heat annealing process consisting of slow heating and cooling within the furnace, in which a large amount of stress is avoided during heating and stress can be released during slow cooling. Therefore, no tensile-strain was introduced into Ir-MnO₂ during the traditional annealing process.

Furthermore, the XRD patterns revealed that the 2θ peaks slightly shifted toward lower angles, and HRTEM images showed that the fringe lattice parameter of the TS-Ir/MnO₂ nanofibers increased by 0.05 Å in comparison with Ir-MnO₂, suggesting that tensile strain existed in TS-Ir/MnO₂ *via* rapid annealing and subsequent cooling.

Moreover, in the EXAFS spectra, which are highly sensitive to local coordination structures, the apparent bond length of the Ir–O first shell in TS-Ir/MnO₂ (1.54 Å) is slightly longer than that of the Ir–O first shell in Ir-MnO₂, clearly demonstrating the tensile strain introduced in TS-Ir/MnO₂ due to rapid annealing-cooling treatment.

Accordingly, in reversed line 5, page 6 of the revised manuscript, the following text **has been added**: “*In this RTAC process, the rapid annealing treatment introduces stress in the lattice, and then rapid cooling does not undergo a slow cooling process for stress release. During this process, the tensile strain was retained in TS-Ir/MnO₂.*”

3. Comment: How to ensure that the Ir atoms are localized at surface Mn sites? Is it displacement doping or atom anchored configuration? This point is very important to understand your OER mechanism.

Reply: We thank the reviewer for the constructive suggestion that is quite useful for improving the quality of this work. The Ir atoms are mainly dispersed on MnO₂ basically in the form of displacement doping. The discussions about the Ir atom configuration are stated as follows.

First, cation exchange is a chemical conversion technique, replacing the cations in a parent ionic material with a different group of cations (Chem. Soc. Rev. 2013, 42, 89; Acc. Chem. Res. 2018, 51, 1711). The progress of a cation exchange reaction may be significantly related to the reduction potential of the cation, which requires the cations in a parent ionic material to have a lower reduction potential (Adv. Mater. 2020, 32, 2001866). Herein, the TS-Ir/MnO₂ preparation was based on a cation exchange method with Ir atoms substituting for surface Mn atoms at first, and then underwent a rapid annealing-cooling treatment process. Note that the higher reduction potential of the Iridium ion ensures that the cation exchange reaction can proceed, suggesting that the Ir atoms substitute for the surface Mn atoms of the MnO₂ nanofiber. Furthermore, the HAADF-STEM images in Fig. 1g and h clearly show Ir atomically dispersed in the obtained TS-Ir/MnO₂ nanofibers. The Ir atoms, highlighted by scattered bright dots in the lattice, are at the same locations as columns of Mn atoms (Fig. 1i), which suggests that atomically dispersed Ir replaces surface Mn sites in the MnO₂ nanofiber lattice *via*

a cation-exchange reaction. Finally, EXAFS spectra can be used to reveal the local coordination environments of Ir species in the samples. In the Fourier-transform (FT) curves of the Ir L_3 -edge EXAFS spectra, the main peaks in the range of 1.5-1.6 Å can be assigned to the first shell of the Ir–O coordination. Another peak at 2.66 Å is shorter than at 3.03 Å (for Ir–Ir bond at IrO₂) and is larger than at 2.45 Å (for Mn–Mn bond at MnO₂), suggesting that the peak can be assigned to Ir–Mn at TS-Ir/MnO₂. Above all, these results clearly demonstrate that atomically dispersed Ir replaces surface Mn sites in the MnO₂ nanofiber lattice *via* a cation-exchange reaction in TS-Ir/MnO₂.

Accordingly, in line 13, page 9 of the revised manuscript, the following text **has been modified**: “Furthermore, another peak at 2.66 Å is shorter than at 3.03 Å (for Ir–Ir bond at IrO₂) and is larger than at 2.45 Å (for Mn–Mn bond at MnO₂), suggesting that the peak can be assigned to Ir–Mn at TS-Ir/MnO₂. This further demonstrates that atomically dispersed Ir replaces surface Mn sites in the MnO₂ nanofiber lattice *via* a cation-exchange reaction in TS-Ir/MnO₂.”

4. Comment: If the Ir atom are doped onto MnO₂ surface, why a larger XRD shift occur at TS-IrMnO₂?

Reply: This is a good question. In the previous question, we have shown that Ir atoms replace surface Mn sites in MnO₂ nanofibers. Although the Ir atoms have a larger atomic radius, we observed no significant XRD shift. This is because the Ir load is only 3.7 wt% and only a few surface Mn atoms are replaced, making it difficult to cause large lattice changes. After the tensile strain is introduced in TS-Ir/MnO₂ *via* a rapid annealing-cooling treatment, the lattice of the entire sample is stretched due to the presence of tensile stress, so that a larger XRD shift occurs at TS-Ir/MnO₂ related to Ir-MnO₂ and MnO₂.

5. Comment: Based on the XAS result, the bond length of Mn-O is hardly changed. This means that the host MnO₂ cannot give rise to a lattice strain. Based on this consideration, a small number of Ir atoms at catalyst surface really can lead to a large XRD difference? It is hard to understand this point.

Reply: We are greatly grateful to the reviewer for your nice question and sorry that the tensile strain and the existence of iridium atoms were not clearly analyzed in the previous manuscript. The tensile strain was introduced to the host MnO₂, leading to a larger bond length of Mn–O bond and a larger XRD difference in TS-Ir/MnO₂ in comparison with Ir-MnO₂. It is discussed in more detail below.

The Ir-MnO₂ was prepared *via* a cation-exchange reaction and then traditional annealing treatment. The XRD pattern (Fig. 2c) shows no significant XRD shift related to MnO₂, which reveals that no tensile stress is introduced into the host MnO₂. Furthermore, as shown in Figure N3, the main peak at 1.47 Å corresponding to the first-shell Mn–O coordination shows no obvious change in comparison with MnO₂, which is primarily due to the low Ir content and no tensile stress in the host MnO₂. For TS-Ir/MnO₂, tensile stress was successfully introduced in the host MnO₂ *via* a rapid annealing-cooling treatment. It could be found that by comparing the Mn–O bond lengths in TS-Ir/MnO₂ *versus* MnO₂, the Mn–O bond lengths increased by 0.03 Å, suggesting that tensile stress existed in the host MnO₂. The larger XRD shift for TS-Ir/MnO₂ further demonstrates the introduction of tensile stress. Above all, based on the XRD and XAFS results, the introduction of low-load Ir to the surface of MnO₂ does not significantly change its lattice, and only when tensile stress is introduced into the host MnO₂ (TS-Ir/MnO₂) *via* a rapid annealing-cooling treatment, it causes the lattice to stretch.

Figure N3. FTs of Mn *K*-edge EXAFS oscillations for TS-Ir/MnO₂, Ir-MnO₂, MnO₂, and Mn foil.

Accordingly, in line 7, page 9 of the revised manuscript, the following text **has been modified**: “*However, the tensile strain effect could be found by comparing the Mn–O bond lengths in TS-Ir/MnO₂ versus MnO₂ (Supplementary Table 1), and the Mn–O bond lengths increased by 0.03 Å, suggesting that tensile stress existed in host MnO₂ for TS-Ir/MnO₂.*”

6. Comment: In fact, any doping strategy all can lead to a change in localized region, which makes it difficult toward to the tensile strain.

Reply: We are greatly grateful to the reviewer for your nice question and it is useful for improving the quality of this work. Heteroatom doping regulates the adsorption strength of reactive species mainly by affecting the local electronic structure, and then significantly improves the performance of electrocatalysts (*Science* 2015, 348, 1230; *Angew. Chem., Int. Ed.* 2020, 59, 4525; *Nat. Commun.* 2023, 14, 345.). The catalytic activity of various types of electrochemical reactions can also be enhanced by strain engineering, usually in the form of compressive or tensile surface strains in carefully engineered core shells or alloyed nanostructures, interface engineering, heat treatment and light treatment (*Science* 2016, 352, 73-76; *Nature* 2019, 574, 81-85; *Adv. Mater.* 2019, 31, e1807001; *Nat. Nanotechnol.* 2021, 16, 1371-1377; *Nat. Energy* 2019, 4, 115-122). In this work, tensile stress is successfully introduced into the host MnO₂ (TS-Ir/MnO₂) *via* a rapid annealing-cooling treatment. Note that the tensile stress is generated by a rapid annealing-cooling treatment process rather than by doping Ir atoms, which can be demonstrated by XRD, TEM and XAFS results in response to the above questions.

The “localized tensile strain” in the previous manuscript is mainly used to emphasize that the L-LOM reaction mechanism at the Ir sites can achieve a fast and continuous OER reaction, which may introduce some ambiguity in that there is only locally generated lattice stress in TS-Ir/MnO₂. In order to avoid ambiguity, in the revised manuscript, we have removed the description of “localized tensile strain”, and the title has been changed to “*Tensile straining of iridium sites in manganese oxides for proton-*

exchange membrane water electrolyzers”.

7. Comment: In Fig. 2, the Ir-O bond in TS-Ir/MnO₂ is stretched comparing to Ir-MnO₂, the relative Mn-O bond should be compressed, especially for neighboring Mn atoms around the doped Ir sites. Why you can observe a contrary behavior? How to understand this reason?

Reply: This is a very good question. X-ray absorption fine structure (XAFS) spectroscopy is an exceptionally powerful characterization technique used to determine the local atomic and electronic structures of materials (Phys. Rev. Lett. 1971, 27, 1204-1207). Because of the strong penetration of hard X-rays, the structural information obtained by XAFS is the average result of the bulk phase of the material (Sci. China Mater. 2015, 58, 313-341). The Ir-O bond in TS-Ir/MnO₂ is stretched compared to Ir-MnO₂ because of the introduction of tensile strain in TS-Ir/MnO₂ *via* a rapid annealing-cooling treatment. Because tensile strain was introduced in the host MnO₂ (TS-Ir/MnO₂), the lattice of the host MnO₂ was stretched. Only a small percentage of surface Mn atoms were replaced by Ir atoms for TS-Ir/MnO₂. Therefore, the average bond length after the combination of a small percentage of compressed M-O bonds and a large percentage of stretched Mn-O bonds is still extended for the original Mn-O bond length (MnO₂ and Ir-MnO₂). This can be proven by the XAFS results of the Mn *K*-edge as shown in Fig. 2e, and the longer Mn-O bonds in TS-Ir/MnO₂ were observed.

Accordingly, in line 7, page 9 of the revised manuscript, the following text **has been modified**: “*However, the tensile strain effect could be found by comparing the Mn-O bond lengths in TS-Ir/MnO₂ versus MnO₂ (Supplementary Table 1), and the Mn-O bond lengths increased by 0.03 Å, suggesting that tensile stress existed in host MnO₂ for TS-Ir/MnO₂.*”

8. Comment: Why the L-LOM cannot occur at Ir-MnO₂? How to understand the contribution of the localized strain onto regulating OER pathway? Some relative calculations can better support your assertions.

Reply: We thank the reviewer for this suggestion. In the lattice oxygen evolution mechanism (LOM), the generated oxygen partially or entirely originates from the lattice

oxygen of the electrocatalyst itself (Nat. Chem. 2017, 9, 457-465). Research has shown that metal oxides with oxygen vacancies, active oxygen atoms and suitable metal-oxygen covalency tend to control the reaction pathway by LOM (Adv. Energy Mater. 2022, 12, 2103670). In this work, after the TS-Ir/MnO₂ sample underwent a rapid annealing-cooling treatment, a certain number of defects and tensile strain were introduced into the sample, which can effectively activate the oxygen atom and regulate the Ir–O covalency, thus realizing the kinetic-fast L-LOM pathway. For comparison, Ir-MnO₂ with low vacancies and inappropriate Ir–O covalency mainly experienced the reaction path of the AEM. This can be further revealed by the SRIR result as shown in Figure N4. Only one absorption band appeared at 1057 cm⁻² with gradually increasing applied potential, suggesting the production of key *OOH species in Ir-MnO₂ under working conditions. The above results clearly demonstrate that Ir-MnO₂ catalyzes the OER, which follows a kinetic-slow AEM pathway.

Figure N4. *In situ* SRIR spectroscopy measurements of Ir-MnO₂ at various potentials

The TS-Ir/MnO₂ catalyzes the OER that follows the L-LOM pathway which can be understood from the following aspects. First, a low-iridium electrocatalyst in which tensile-strained iridium atoms are localized at manganese-oxide surface cation sites (TS-Ir/MnO₂) was successfully synthesized, where the strong interaction between metal atoms and support optimizes the covalency of Ir–O bonds and can profoundly determine the catalytic OER performance and reaction mechanism. The TEM and

XAFS results clearly demonstrate the presence of tensile strain that can efficiently regulate the covalency of Ir–O bonds in TS-Ir/MnO₂ to realize a high oxidation state of Ir sites, which can accelerate the reaction kinetics. Then, the correlation of multiple *in situ* characterization techniques contributes to unraveling the unique reaction mechanisms and the lifetime structure-performance relationships (Acc. Chem. Res. 2022, 55, 1949-1959). According to *in situ* SRIR spectra, only key O–O intermediates rapidly accumulated over the active Ir sites in TS-Ir/MnO₂. For comparison, the *OOH species were observed for IrO₂ and Ir-MnO₂ during the OER process. Furthermore, isotope-labeling *in situ* SRIR spectra revealed that ¹⁶O–¹⁸O radicals are derived from adsorbed H₂O molecules and lattice oxygen atoms for TS-Ir/MnO₂, suggesting a LOM-like mechanism on the TS-Ir/MnO₂ catalyst surface. In the *in situ* XAFS results, the local coordination structure of Ir sites first increased and then decreased when the potentials increased from 1.30 to 1.43 V *versus* RHE, implying that a potential-driven structural evolution occurs during the OER by releasing lattice oxygen atoms from TS-Ir/MnO₂ localized surface sites to induce the localized LOM (L-LOM) reaction mechanism. Clearly, tensile strain introduced in TS-Ir/MnO₂ can enhance the covalency of the Ir–O bond to improve the deprotonation ability and increase the Ir oxidation state, triggering a new continuous localized lattice oxygen-mediated (L-LOM) mechanism under working conditions. More interestingly, the tensile strain localized on the MnO₂ surface could tailor the adsorption behavior of Ir sites to accelerate the deprotonation of *OH at surface oxygen vacancies and effectively prevent the local peroxidation of Ir sites to reduce the dissolution and maintain the structural integrity of the catalyst. Above all, the reaction mechanism of different samples was uncovered by the *in situ* XAFS and SRIR results, and a continuous kinetic-fast L-LOM mechanism was realized for the TS-Ir/MnO₂ during the OER process.

Accordingly, Figure N4 has been added in the Supplementary Information as Supplementary Fig. 26. In line 10, page 17 of the revised manuscript, the following text **has been modified**: “Above all, tensile strain introduced in TS-Ir/MnO₂ can enhance the covalency of the Ir–O bond to improve the deprotonation ability and increase the Ir

oxidation state, triggering a new continuous localized L-LOM mechanism under working conditions.” In line 2, page 14 of the revised manuscript, the following text **has been added**: “The SRIR result of Ir-MnO₂ further shows that the production of key *OOH species under working conditions, suggesting that the Ir-MnO₂ catalyses the OER that follows a kinetic-slow AEM pathway (Supplementary Fig. 26).”

9. Comment: Why the I-t curves in Fig. 3f is slowly decreased? But the V-t curves in Fig. 6c are slowly increased.

Reply: We thank the reviewer for this nice question. The operation durability is very important for practical applications of electrocatalysts. To evaluate the operation durability of the catalyst, chronoamperometry (I-t curves) at a constant potential of 1.60 V (driving a current density of 200 mA cm⁻²) was performed, as shown in Fig. 3f. TS-Ir/MnO₂ still maintains a satisfactory ~93% of the initial current density after 100 h of operation. The slow decrease of I-t curves in Fig. 3f means a slight decline in stability because of the slight corrosion of the carbon clothes substrate and a lot of bubbles produced under a high current density. Furthermore, to verify the potential of TS-Ir/MnO₂ for industrial applications, we constructed a PEMWE device using a NafionTM 117 membrane, and TS-Ir/MnO₂ and commercial Pt/C worked as anode and cathode catalysts, respectively, in an acidic electrolyte. As shown in Fig. 6c, the resultant electrolyzer maintains a current density of 500 mA cm⁻² at an applied bias of ~1.75 V and can continuously operate for 200 h with negligible attenuation under simulated industrial conditions (at 80 °C). Note that the V-t curves in Fig. 6c are slowly increased, suggesting that a slightly larger voltage is needed to achieve a current density of 500 mA cm⁻². This also means a slight decline in stability under 500 mA cm⁻². Therefore, both the slow decrease of the I-t curves in Fig. 3f (chronoamperometry) and the slow increase of the V-t curves in Fig. 6c (chronopotentiometry) represent a slight decline in stability at large current densities under acidic environments.

Accordingly, in line 3, page 10 of the revised manuscript, the following text **has been modified**: “OER stability tests and constant S-numbers calculated at a constant potential of

1.60 V (driving a current density of 200 mA cm⁻²) for TS-Ir/MnO₂.”

10. Comment: How to estimate the number of the active sites in different catalyst? In other words, how to ensure the equivalent Ir atom at MnO₂ surface between Ir-MnO₂ and TS-Ir/MnO₂.

Reply: We thank the reviewer for the constructive suggestion that is quite useful for improving the consistency of this work and we also compliment the reviewer for your expert knowledge in the field of electrochemistry. We are sorry that the calculation method of the number of active sites was not clearly described in the previous manuscript.

The catalytic intrinsic activity, such as mass activity and TOF, is obtained according to the active sites. Single-atom catalysts have the characteristics of uniformly dispersed active sites, high atomic utilization and accessible metal centers, and its single atom is considered as the catalytic active center, so the number of catalytic active centers can be calculated according to the single atom load (Nat. Commun. 2022, 13, 3822; Nat. Commun. 2019, 10, 4849). In this research system, the electrochemical characterization results obviously reveal that the performance parameters of the TS-Ir/MnO₂ catalyst, including the overpotential and Tafel slope, are superior to those of commercial IrO₂ and MnO₂. This means that the catalytic activity of the TS-Ir/MnO₂ catalyst mainly comes from the metal Ir sites. Therefore, the number of active sites of TS-Ir/MnO₂ and Ir-MnO₂ catalysts can be obtained by generally regarding the metal Ir sites as the active sites. First, the metal content of the TS-Ir/MnO₂ catalyst was calculated from the Ir loading mass on the host MnO₂, which was 3.9 wt% as verified by the inductively coupled plasma optical emission spectrometry (ICP-OES) results. The raw material feeding ratio of TS-Ir/MnO₂ catalyst and Ir-MnO₂ in the early preparation stage is the same, which can ensure that the Ir loading of the two samples is basically similar. The high exposure of metal-active sites on the MnO₂ surface enables them to participate in OER reactions. Therefore, the number of the active sites can be obtained from the single atom Ir load in TS-Ir/MnO₂ and Ir-MnO₂, and they show similar numbers of active sites.

Accordingly, in line 2, page 7 of the revised manuscript, the following text **has been**

modified: *“Furthermore, the ICP–OES result shows that the Ir loading is similar in Ir-MnO₂ attributed to the same raw material feeding.”*

11. Comment: In your OER test, some Ir atoms are partial dissociated, which may generate a dynamic balance to affect the catalytic reactivity (Nature Catalysis, 4, 1012-1023, 2021). How to exclude this factor?

Reply: Thank you for your constructive suggestion and it is useful for improving the quality of this work. According to your suggestion, we performed ICP-OES characterization of the electrolyte at different reaction times to observe the Ir quality. The Ir quality underwent a slow increase and no decreasing trend was observed during the first few hours of OER operation, suggesting that the dissolved Ir did not have an obvious dynamic sediment-dissolution equilibrium. Therefore, the catalytic activity mainly comes from the continuous localized lattice oxygen-mediated (L-LOM) pathway in TS-Ir/MnO₂.

Recently, Lee’s group reported that a highly loaded Ru atom (12 wt%) catalyst can carry out a dynamic Ru dissolving-redeposition process at the initial stage of the electrochemical reaction, which is conducive to the formation of the Ru array active sites, and then improves the catalyst activity. It may occur in a system of highly loaded, regularly arranged single-atom Ru catalysts (Nat. Catalysis, 2021, 4, 1012-1023). In this work, we synthesized a dispersible single-atom Ir catalyst with a low metal loading of 3.9 wt%, and no Ir array sites were observed (TEM image in Fig. 1g). Furthermore, to further determine whether the dissolving-redeposition phenomenon of Ir atoms occurs in electrochemical tests, the metal concentration in the electrolyte was analyzed by ICP-OES. Figure N5 shows the time dependence of the Ir quality when the electrolysis current density was set at 200 mA cm⁻². The Ir quality underwent a slow increase and no decreasing trend was observed during the first 40 hours of OER operation, suggesting that the dissolved Ir did not have an obvious dynamic sediment-dissolution equilibrium. Besides, no significant Ir array sites were observed in the TEM image after the reaction. These results clearly demonstrate that the dissolved Ir did not

have an obvious dynamic sediment-dissolution equilibrium during the OER process, suggesting a significant effect on the catalytic performance of dissolved Ir.

Accordingly, Figure N5 has been added in the Supplementary Information as Supplementary Fig. 21a. In line 10, page 12 of the revised manuscript, the following text **has been added**: “*The Ir concentration underwent a slow increase and no decreasing trend was observed during the first few hours of OER operation, suggesting that the dissolved Ir did not have an obvious dynamic sediment-dissolution equilibrium.*”

Figure N5. Dissolved content of metal Ir for TS-Ir/MnO₂ during OER process.

Reply to Reviewer #3:

The manuscript by Su et al. designed a low-iridium electrocatalyst in which tensile-strained iridium atoms were introduced and localized at the manganese-oxide surface cation sites (TS-Ir/MnO₂) for efficient and stable acidic-oxygen evolution reaction (OER) activity. The choice of methods is appropriate, and the authors performed controlled experiments such as in situ X-ray absorption fine structure (XAFS) and synchrotron radiation infrared (SR-IR) spectroscopy at the OER working conditions for investigating the dynamic catalytic process and revealing the catalytic reaction mechanism of the catalyst. What impressed me the most was the authors' observation of a new localized lattice oxygen-mediated (L-LOM) mechanism in the well-designed

TS-Ir/MnO₂ catalyst that can regulate the kinetics of acidic-OER and optimize the structural integrity of the catalyst during the reaction process, which is likely to be of wide interest to the catalytic mechanism understanding and catalyst design for the new energy technologies such as the proton-exchange membrane water electrolysis. Overall, I think that the whole work is of novelty and the findings are of particular significance. Therefore, I recommend the publication of this manuscript in the high-profile journal of Nature Communications after the following issues are clarified:

We are very grateful to Reviewer #3 for highlighting the significance and novelty of this work. According to your suggestion, we have carefully added performance characterization and related analysis to meet the high requirements of *Nature Communications*.

1. Comment: In the experimental and catalyst preparation sections, the authors claimed that tensile strain was introduced by a rapid thermal annealing–cooling strategy. As is known, the temperature of annealing will affect the structure of samples and then affect the performance of the samples. I wonder if the authors tried other temperatures, such as the lower or higher temperatures than that of used by the authors in the manuscript?

Reply: Thank you for your careful consideration. Specifically, the target sample is rapidly heated to 250 °C for 2 h and then cooled rapidly to obtain the TS-Ir/MnO₂ electrocatalyst. At 250 °C, the tensile strain was successfully introduced into the sample and the Ir atoms were uniformly distributed in the MnO₂ substrate with suitable Ir–O covalency for efficient OER activity.

Noticeably, a rapid thermal annealing-cooling process could introduce the strain effect in the samples, and then subsequently influences the catalytic activity (Nat. Nanotechnol. 2021, 16, 1371-1377). The substrate has a great influence on the properties, and the morphology and structure will change obviously with the increase of temperature. However, the disadvantages of higher temperature are the collapse of substrate morphology and the agglomeration of single atoms on the surface, thus reducing the catalytic activity (Adv. Energy Mater. 2018, 8, 1702476). Considering that the thermal annealing temperature will affect the strain degree and the distribution of Ir,

the samples were also performed at different pyrolysis temperatures (150 °C and 350 °C). As shown in the Figure N6, the as-synthesized 150 °C sample (thermal annealing temperature of 150 °C) has clear nanofibers structure similar to TS-MnO₂ without nanoparticles and delivers an overpotential of 242 mV at 10 mA cm⁻², which reveals unsuitable Ir–O covalency attributed to the inadequate strain effect at a lower temperature. Furthermore, Figure N6d clearly shows that the small Ir nanoparticles were distributed on the MnO₂ substrate and a higher overpotential of 271 mV at 10 mA cm⁻² was observed for the as-synthesized 350 °C sample (thermal annealing of 350 °C), suggesting that Ir atoms aggregated into nanoparticles at a higher temperature. Above all, considering the strain effect and the phase composition of Ir sites, the thermal annealing temperature of 250 °C was selected for the TS-Ir/MnO₂ electrocatalyst. This can ensure that the tensile strain was successfully introduced into the sample and that the Ir atoms were uniformly distributed in the MnO₂ substrate with suitable Ir–O covalency for an efficient 4-electron OER process.

Figure N6. (a) Linear sweep voltammetry (LSV) curves for Ir electrocatalysts thermal annealed at 150, 250 and 350 °C. TEM images of Ir-based electrocatalysts annealed at (b) 250 °C, (c) 150 °C and (d) 350 °C.

Accordingly, Figure N6 has been added in the Supplementary Information as Supplementary Fig. 14. In reversed line 13, page 10 of the revised manuscript, the following text **has been added:** “*The morphology structure and OER performance of Ir electrocatalysts at different annealing pyrolysis temperatures are shown in Supplementary Fig. 14, revealing the optimal tensile strain and atomically dispersed Ir active sites in the TS-Ir/MnO₂ (250 °C) electrocatalyst.*”

2. Comment: In the main body of the text, the authors mentioned that the localized lattice oxygen-mediated (L-LOM) mechanism was realized for TS-Ir/MnO₂. To consolidate this conclusion, the authors should strengthen the relationship between the structure (such as the atomic and/or electronic structure of the catalyst) and reaction mechanism.

Reply: We thank the reviewer for the nice question. We are sorry that the causal relationship between structure and reaction mechanism was not clearly expressed in the previous manuscript. These issues would be further discussed in detail in the revised manuscript.

Firstly, a low-iridium electrocatalyst in which tensile-strained iridium atoms are localized at manganese-oxide surface cation sites (TS-Ir/MnO₂) was successfully synthesized, where the strong interaction between metal atoms and support optimizes the covalency of Ir–O bonds and can profoundly determine the catalytic OER performance and reaction mechanism. The TEM and XAFS results clearly demonstrate the presence of tensile strain that can efficiently regulate the covalency of Ir–O bonds in TS-Ir/MnO₂ to realize a high oxidation state of Ir sites, which can accelerate the reaction kinetics. Then, the correlation of multiple *in situ* characterization techniques contributes to unraveling the unique reaction mechanisms and the lifetime structure-performance relationships (Acc. Chem. Res. 2022, 55, 1949-1959). According to *in situ* SRIR spectra, only key O–O intermediates rapidly accumulated over the active Ir sites

in TS-Ir/MnO₂. For comparison, the *OOH species were observed for IrO₂ and Ir-MnO₂ during the OER process. Furthermore, isotope-labeling *in situ* SRIR spectra revealed that ¹⁶O–¹⁸O radicals are derived from adsorbed H₂O molecules and lattice oxygen atoms for TS-Ir/MnO₂, suggesting a LOM-like mechanism on the TS-Ir/MnO₂ catalyst surface. In the *in situ* XAFS results, the local coordination structure of Ir sites first increased and then decreased when the potentials increased from 1.30 to 1.43 V *versus* RHE, implying that a potential-driven structural evolution occurs during the OER by releasing lattice oxygen atoms from TS-Ir/MnO₂ localized surface sites to induce the localized LOM (L-LOM) reaction mechanism. Clearly, the tensile strain introduced in TS-Ir/MnO₂ can enhance the covalency of the Ir–O bond to improve the deprotonation ability and increase the Ir oxidation state, triggering a new continuous localized lattice oxygen-mediated (L-LOM) mechanism under working conditions. More interestingly, the tensile strain localized on the MnO₂ surface could tailor the adsorption behavior of Ir sites to accelerate the deprotonation of *OH at surface oxygen vacancies and effectively prevent the local peroxidation of Ir sites to reduce the dissolution and maintain the structural integrity of the catalyst.

Accordingly, in line 10, page 17 of the revised manuscript, the following text **has been modified**: “*Above all, tensile strain introduced in TS-Ir/MnO₂ can enhance the covalency of the Ir–O bond to improve the deprotonation ability and increase the Ir oxidation state, triggering a new continuous localized L-LOM mechanism under working conditions.*”

3. Comment: In the electrocatalytic performance characterizations, it is known that the environment of electrochemical testing has a certain effect on the performance of the catalyst. In this regard, I suggest that the potential temperature effects should be considered and some related discussions on this issue should be included.

Reply: Thank you for your nice questions. We really praise the reviewer for your expert knowledge in the field of electrochemistry. According to your suggestion, the electrochemical measurements were performed at different temperatures to determine

the relationship between the working temperature and activity. Seen from Figure N7, it can be drawn that the TS-Ir/MnO₂ electrocatalyst delivers faster kinetics and better activity at a higher temperature.

The temperature of the electrochemical cell is an important parameter that significantly affects the electrochemical performance tests (*J. Am. Chem. Soc.* 140, 2926-2932 (2018)). Generally, the performance tests of the electrochemical cell were carried out at room temperature. In order to further explore the temperature effect, the electrochemical cell was put on a temperature-control heater with the temperature controlled at 25, 50 and 80 °C for temperature-dependent studies. As shown in Figure N7, the OER activity is quite temperature-dependent, and the overpotentials of TS-Ir/MnO₂ electrocatalyst decrease from 198 to 180 mV at a current density of 10 mA cm⁻² with the increase of temperature to 80 °C, which is consistent with what has been observed that a higher temperature provides faster OER kinetics and better OER activity (*J. Power Sources* 167, 235-242 (2007)). Based on the above results, the TS-Ir/MnO₂ electrocatalyst shows temperature-dependent OER activity with faster OER kinetics at higher temperatures.

Figure N7. Linear sweep voltammetry (LSV) curves of TS-Ir/MnO₂ under 25, 50 and 80 °C in 0.1 M HClO₄.

Accordingly, Figure N7 has been added in the Supplementary Information as Supplementary Fig. 15. In reversed line 9, page 10 of the revised manuscript, the

following text **has been added**: “Considering the effect of temperature on acidic OER performance, we conducted a water splitting test under different operating temperatures. The OER activity is quite temperature dependent, and the overpotentials of the TS-Ir/MnO₂ electrocatalyst decrease from 198 to 180 mV at a current density of 10 mA cm⁻² with the increase of temperature to 80 °C, which is consistent with what has been observed that a higher temperature provides faster OER kinetics and better OER activity (Supplementary Fig. 15).”

4. Comment: The stability of catalysts is more important than the catalytic activity for the practical applications. A suitable explanation for the observed stability of the catalyst would open a great avenue for further developments of acidic-OER catalysts. Therefore, I strongly suggest the authors strengthening related discussions on this issue.

Reply: We are greatly grateful to the reviewer for your nice question and it is useful for improving the quality of this work. According to your suggestion, the stability of the acidic-OER for the TS-Ir/MnO₂ electrocatalyst will be analyzed in detail from three aspects: structure, mechanism and performance.

The operation stability is an important index to evaluate the performance of electrocatalysts, especially for the acidic OER. It is known that for highly reactive iridium electrocatalysts, especially for commercial IrO₂, the high oxidation voltage causes an excessive increase of the oxidation state and a decrease of crystallization of the electrocatalyst (*J. Am. Chem. Soc.* 139, 2017, 12837-12846; *J. Electroanal. Chem.* 2016, 774, 102-110). Additionally, a number of recent studies have unveiled the *in-situ* reconstruction of electrocatalyst surfaces, which can in turn affect their intrinsic OER-activity due to a significant change of the surface oxidation state as well as agglomeration and dissolution of surface-active phases (*Nat. Catal.* 2018, 1, 300-305; *Adv. Mater.* 2018, 30, 1804333). To improve the durability of the electrocatalyst under acidic OER operation conditions, the key is to construct a stable coordination structure of Ir active sites on a steady substrate to inhibit the peroxidation and the aggregation and dissolution of the surface-active phase under high potentials.

Structure: In this work, we selected MnO₂, which has a stable structure under acidic

oxidation conditions, as the carrier material (Angew. Chem. 2019, 131, 5108-5112; Nat. Catal. 2021, 4, 1012-1023). A low-iridium electrocatalyst in which atomically dispersed and tensile-strained iridium sites were confined in manganese oxides (TS-Ir/MnO₂) was hydrothermally synthesized and then rapidly thermally annealed and subsequently cooled, which is hereafter called the “rapid thermal annealing-cooling” (RTAC) strategy. It is noted that atomically dispersed Ir active sites with stable Ir–O moieties were strongly coupled on the lattice of the MnO₂ substrate. This stable configuration has the potential to inhibit the peroxidation and aggregation of Ir active sites at high potentials.

Mechanism: The *in situ* SRIR and XAFS techniques reveal that tensile strain introduced in TS-Ir/MnO₂ can enhance the covalency of the Ir–O bond to improve the deprotonation ability and increase the Ir oxidation state, triggering a new continuous localized lattice oxygen-mediated (L-LOM) mechanism under working conditions. More interestingly, the tensile strain localized on the MnO₂ surface could tailor the adsorption behavior of Ir sites to accelerate the deprotonation of *OH at surface oxygen vacancies and effectively prevent the local peroxidation of Ir sites to reduce the dissolution and maintain the structural integrity of the catalyst for high activity and durability.

Performance: TS-Ir/MnO₂ presents good catalytic stability at current densities of 10 and 200 mA cm⁻² after 200 and 100 h of continuous OER tests with negligible performance degradation. More importantly, the resultant PEM electrolyzer delivered a current density of 500 mA cm⁻² at an applied bias of ~1.75 V for over 200 h under simulated industrial conditions. Meanwhile, the morphology characterization showed that no collapse of the MnO₂ substrate and no obvious particles were observed after OER tests. Above all, the TS-Ir/MnO₂ electrocatalyst with atomically dispersed Ir-O₄ moieties delivers high durability due to the stable structure and a continuous L-LOM mechanism.

Accordingly, in reversed line 4, page 12 of the revised manuscript, the following text **has been added:** “*In summary, the high durability of the TS-Ir/MnO₂ electrocatalyst under continuous operating conditions is attributed to the stable structure and a*

continuous L-LOM mechanism during the OER process, which can inhibit peroxidation and dissolution of Ir active sites.”

5. Comment: General comment. Some of the potentials listed throughout the manuscript need to be listed/provided relative to a reference electrode.

Reply: Thank you for your careful inspection. According to your suggestion, some of the potentials listed throughout the manuscript have been listed relative to a reference hydrogen electrode (vs. RHE). Besides, we checked the full manuscript carefully to ensure that there were no similar mistakes.

6. Comment: There are some grammatical and handwriting errors, in the page of 7 “which reveals the electron transfer from Mn to Ir.”, “the main peaks in the range 1.5-1.6 Å” and “for which the corresponding data are listed in Table 1”. In the page of 16 “...TS-Ir/MnO₂ and commercial Pt/C as anode and cathode catalysts”. In Supplementary Information, Fig. 12 “The corresponding overpotentials at current density of 10, 100 and 200 mA cm⁻²”, Fig. 13 “OER polarization curves of TS-Ir/MnO₂ of different metal loading of Ir” and Fig. 21 “(b) O 1s XPS spectra for TS-Ir/MnO₂ after electrochemical measurements”. The authors should check them out carefully.

Reply: Thank you for your careful inspection. According to your suggestions, “which reveals the electron transfer from Mn to Ir.” **has been changed** in “which reveals the electron transfer from Ir to Mn.”; “the main peaks in the range 1.5-1.6 Å” **has been changed** in “the main peaks in the range of 1.5-1.6 Å”; “for which the corresponding data are listed in Table 1” **has been changed** in “for which the corresponding data are listed in Supplementary Table 1”; “...TS-Ir/MnO₂ and commercial Pt/C as anode and cathode catalysts” **has been changed** in “...TS-Ir/MnO₂ and commercial Pt/Ti as anode and cathode catalysts”; “The corresponding overpotentials at current densities of 10, 100 and 200 mA cm⁻²” **has been changed** in “The corresponding overpotentials at current densities of 10, 100 and 200 mA cm⁻²”; “OER polarization curves of TS-Ir/MnO₂ with different metal loadings of Ir” **has been changed** in “OER polarization

curves of TS-Ir/MnO₂ with different metal loadings of Ir”; “(b) O 1s XPS spectra for TS-Ir/MnO₂ after electrochemical measurements” **has been removed** in the revised manuscript. Besides, we checked the full manuscript carefully to ensure that there were no similar mistakes.

REVIEWER COMMENTS

Reviewer #1 (Remarks to the Author):

I appreciate that the authors have done a good job in addressing my comments connected to the earlier version of the manuscript. All the technical issues have been taken care of in a satisfactory way. I would support publication at this point.

Reviewer #2 (Remarks to the Author):

Most of problems have been addressed and these experimental evidences are very powerful to support the assertions of the authors. However, some minor points need to be further considered before acceptance.

(1) In your response, the changes in XRD are attributed to the lattice strain. Is that related with doping Ir atom? The XRD shift of the MnO₂ sample with different doping concentration and annealing method need to be compared that may be a good way to support your experiments.

(2) In addition, in your discussion about Ir-doped MnO₂, the ion exchange process also can be affected by light irradiation, such as *Angew. Chem. Int. Ed.*, 62, e202301128 (2023), which can directly affect the distribution of Ir atoms. The atomic configuration difference between this reference and your report should be compared and discussed.

Reviewer #3 (Remarks to the Author):

The comments were well addressed, and the current manuscript can be accepted.

Response to Reviewers' Comments

We are grateful to the reviewers for having given us important and valuable comments on the manuscript NCOMMS-23-39607A entitled: “*Tensile straining of iridium sites in manganese oxides for proton-exchange membrane water electrolyzers*”. The detailed replies to your comments are presented in a point-to-point manner as follows. **The modifications in the revised manuscript are highlighted in the yellow background.**

Reviewer #1 (Remarks to the Author):

I appreciate that the authors have done a good job in addressing my comments connected to the earlier version of the manuscript. All the technical issues have been taken care of in a satisfactory way. I would support publication at this point.

Reply: Thank you very much for your approval and guidance on our manuscript.

Reviewer #2 (Remarks to the Author):

Most of problems have been addressed and these experimental evidences are very powerful to support the assertions of the authors. However, some minor points need to be further considered before acceptance.

(1) In your response, the changes in XRD are attributed to the lattice strain. Is that related with doping Ir atom? The XRD shift of the MnO₂ sample with different doping concentration and annealing method need to be compared that may be a good way to support your experiments.

Reply: This is a very nice question. According to your suggestion, the XRD shifts of the MnO₂ samples with different doping concentrations and annealing methods are provided and analyzed in the revised manuscript.

The XRD results (Fig. 1c and Supplementary Fig. 2) show that a larger XRD shift occurs at TS-Ir/MnO₂ related to Ir-MnO₂ because of the presence of tensile stress after a rapid annealing-cooling treatment. Furthermore, the doping Ir atoms have no significant effect on the XRD shift in this manuscript because the Ir load is only 3.9 wt% and only a few surface Mn atoms are replaced, making it difficult to cause large lattice changes. Furthermore, to further verify the effect of doping Ir atoms and the annealing

method on the XRD results, different Ir doping concentrations (2.5 wt%, 3.9 wt%, 5.1 wt%) and annealing methods (rapid annealing-cooling treatment and traditional annealing treatment) were applied as shown in Figure N1. A larger XRD shift is observed for the TS-Ir/MnO₂ and MnO₂ (annealing-cooling) related to Ir-MnO₂ and MnO₂, which reveals that a rapid annealing-cooling treatment introduces tensile strain in the host MnO₂ lattice, resulting in a shift in XRD. Note that the XRD diffraction peaks with different Ir doping concentrations do not move significantly with the increase of Ir loading, except for a slight low-angle shift and a weak IrO₂ diffraction peak in the high Ir loading sample (5.1 wt%), which indicates that low Ir loading will not cause obvious lattice expansion in the host MnO₂ in this manuscript.

Figure N1. (a) and (b) XRD patterns for TS-Ir/MnO₂, Ir-MnO₂ and MnO₂ electrocatalysts. (c) and (d) XRD patterns for MnO₂, Ir-MnO₂ (2.5 wt%, 3.9 wt% and 5.1 wt%) and TS-Ir/MnO₂ electrocatalysts.

Accordingly, Supplementary Fig. 2 has been replaced by Figure N1 in the revised Supplementary Information, and in line 9, page 6 of the revised manuscript, the

following text **has been modified**: “Furthermore, the XRD results of the Ir samples with different Ir doping concentrations and annealing methods clearly reveal that a rapid annealing-cooling treatment introduces tensile strain in the host MnO₂ lattice resulting in a larger shift in the XRD pattern, excluding the effect of low loading Ir atoms on the lattice of host MnO₂.”

(2) In addition, in your discussion about Ir-doped MnO₂, the ion exchange process also can be affected by light irradiation, such as Angew. Chem. Int. Ed, 62, e202301128 (2023), which can directly affect the distribution of Ir atoms. The atomic configuration difference between this reference and your report should be compared and discussed.

Reply: Thank you for your insightful comment and constructive suggestion, which is quite useful for improving the quality of this work. Recently, Liu’ group presented a light-driven strategy to realize an orderly Ir atomic assembly on a F doped MnO₂ (IrMnOF) surface in which the introduced F effectively softens the bonding strength of Mn–O bonds to induce an orderly cationic replacement by Ir atoms, leading to the formation of Ir-atomic chains (Angew. Chem. Int. Ed, 2023, 62, e202301128). The TEM and aberration-corrected HAADF-STEM images reveal that Ir is mainly distributed on the host MnOF carrier in the form of orderly arranged Ir-atomic chains and a small number of Ir clusters. In Liu’s work, the orderly arranged Ir-atomic chains with a spin-related lower entropy can be correspondingly optimized to reduce the intrinsic activation energy at potential-determining intermediates for a stable OER process. In our work, we used cation exchange and a subsequent rapid annealing-cooling strategy to prepare a low-iridium electrocatalyst in which iridium atoms are localized at the surface Mn sites of tensile-strained MnO₂. The HAADF-STEM images and XAFS results clearly show that atomically dispersed Ir replaces surface Mn sites. There is no formation of orderly arranged Ir-atomic chains and the presence of adjacent Ir–O–Ir, suggesting uniformly distributed single atomic Ir sites. Meanwhile, benefitting from the introduction of tensile strain *via* a rapid annealing-cooling strategy, the well-designed TS-Ir/MnO₂ electrocatalyst can trigger a new continuous localized lattice oxygen-mediated (L-LOM) mechanism under working conditions identified by *in situ*

synchrotron characterizations. In particular, the L-LOM process could substantially boost the adsorption and transformation of H₂O molecules over the oxygen vacancies around the tensile-strained Ir sites and prevent further loss of lattice oxygen atoms in the inner MnO₂ bulk to optimize the structural integrity of the catalyst.

Accordingly, in line 12, page 3 of the revised manuscript, the following text **has been modified**: “*Liu’s group presented a light-driven strategy to realize an orderly Ir atomic assembly on a F doped MnO₂ (IrMnOF) surface with a spin-related lower entropy that is optimized to reduce the intrinsic activation energy at potential-determining intermediates for a stable OER process.*”

Reviewer #3 (Remarks to the Author):

The comments were well addressed, and the current manuscript can be accepted.

Reply: Thank you very much for your approval and guidance on our manuscript.